# Glioma-initiating cells at tumor edge gain signals from tumor core cells to promote their malignancy

Soniya Bastola[1,16], Marat S. Pavlyukov[2,16], Daisuke Yamashita [1], Sadashib Ghosh[1], Heejin Cho[3,4], Noritaka Kagaya [5], Zhuo Zhang[6], Mutsuko Minata[1], Yeri Lee[3,4], Hirokazu Sadahiro[7], Shinobu Yamaguchi[1], Svetlana Komarova[1], Eddy Yang [6], James Markert[1], Louis B. Nabors [8], Krishna Bhat[9], James Lee[10], Qin Chen[1,11], David K. Crossman [12], Kazuo Shin-Ya [5], Do-Hyun Nam[13,14] & Ichiro Nakano [1,15] ✉

Intratumor spatial heterogeneity facilitates therapeutic resistance in glioblastoma (GBM). Nonetheless, understanding of GBM heterogeneity is largely limited to the surgically resectable tumor core lesion while the seeds for recurrence reside in the unresectable tumor edge. In this study, stratification of GBM to core and edge demonstrates clinically relevant surgical sequelae. We establish regionally derived models of GBM edge and core that retain their spatial identity in a cell autonomous manner. Upon xenotransplantation, edge-derived cells show a higher capacity for infiltrative growth, while core cells demonstrate core lesions with greater therapy resistance. Investigation of intercellular signaling between these two tumor populations uncovers the paracrine crosstalk from tumor core that promotes malignancy and therapy resistance of edge cells. These phenotypic alterations are initiated by HDAC1 in GBM core cells which subsequently affect edge cells by secreting the soluble form of CD109 protein. Our data reveal the role of intracellular communication between regionally different populations of GBM cells in tumor recurrence.

[1] Department of Neurosurgery, University of Alabama at Birmingham, Birmingham, AL 35294, USA. [2] Shemyakin-Ovchinnikov Institute of Bioorganic Chemistry, Moscow 117997, Russian Federation. [3] Research Institute for Future Medicine, Seoul 06351, Republic of Korea. [4] Institute for Refractory Cancer Research, Samsung Medical Center, Sungkyunkwan University School of Medicine, Seoul 06351, Republic of Korea. [5] Biomedical Information Research Center, National Institute of Advanced Industrial Science and Technology, 2-4-7 Aomi, Koto-ku, Tokyo 135-0064, Japan. [6] Department of Radiation Oncology, University of Alabama at Birmingham, Birmingham, AL 35233, USA. [7] Department of Neurosurgery, Yamaguchi University, Yamaguchi, Japan. [8] Department of Neurology, University of Alabama at Birmingham, Birmingham, AL 35294, USA. [9] Department of Translational Molecular Pathology and Brain Tumor Center, The University of Texas, M.D. Anderson Cancer Center, Houston, TX 77030, USA. [10] Department of Chemical and Biomolecular Engineering, Ohio State University, Columbus, OH 43210, USA. [11] Department of Integrative medicine, Tongji Hospital, Tongji Medical College, Huazhong University of Science and Technology, Wuhan 430030, People's Republic of China. [12] Department of Genetics, University of Alabama at Birmingham, Birmingham, AL 35294, USA. [13] Department of Neurosurgery, Samsung Medical Center, Sungkyunkwan University School of Medicine, Seoul 06351, Republic of Korea. [14] Department of Health Science and Technology, Samsung Advanced Institute for Health Science and Technology, Sungkyunkwan University, Seoul 06351, Republic of Korea. [15] Research and Development Center for Precision Medicine, Tsukuba University, Tsukuba, Japan. [16]These authors contributed equally: Soniya Bastola, Marat S. Pavlyukov. ✉email: ichironakano1369@gmail.com

Glioblastoma (GBM) is the most common primary brain tumor and kills more than half of patients within 2 years despite multi-modal approaches involving maximal safe surgical resection, followed by radiation, and chemotherapy[1,2]. Curative efforts by the primary treatment modality, surgery, have been confounded by GBM's highly infiltrative nature[3]—a phenotypic property contributing to unfavorable outcome in many carcinomas[4].

A characteristic feature of GBM is high degree of intratumoral heterogeneity. Jin et al.[5] addressed spatial cellular heterogeneity in tumor core combined with transcriptional subtyping and observed that the peripheral portion of tumor core preferentially express the proneural (PN) genes, which is resected by surgery along with the central core portion expressing the mesenchymal (MES) genes, raising a question for clinical significance of this data. Puchalski et al.[6] subsequently presented the Ivy Glioblastoma Atlas Project (IvyGAP), an anatomical atlas of human GBM that contains mutation and gene expression data obtained from morphologically distinct regions within the tumors. In this IvyGAP dataset, different regions of GBM were primarily identified by the histological appearance of the tumor sections, due to the fact that only a limited number of protein markers for GBM spatial heterogeneity have been identified so far. These include the cell surface markers, CD133 and CD109. CD133 (Prominin-1), a glycoprotein expressed on neural precursor cells, is a well-known marker for identification of the glioma-initiating cell subpopulation among others[7]. Cell surface expression of CD133 appears to be correlated with tumor-initiating properties following current first-line post-surgical therapies[8,9]. Of note, CD133-positive cells appeared to be enriched in the infiltrating edge of GBM tumors[10]. On the other hand, CD109, a glycosylphosphatidylinositol-anchored glycoprotein, is highly expressed in multiple tumors and associated with worse outcome of patients. Recently, we demonstrated an association of CD109 with the tumor-initiating population located at the tumor core in glioma[11]. However, the applicability of these two markers for spatial identity has not been thoroughly validated.

These aforementioned studies primarily describe GBM heterogeneity within the resectable region of the tumor, while the functional characterization of tumor cells at the infiltrating edge has largely remain elusive. This is due to the presence of functional normal brain tissues in the peritumoral edge lesion, leading to surgical inaccessibility for isolation and characterization of the tumor cells located therein. Hence, there is no doubt that these elusive cells contain seeds for fatal GBM recurrence in patients. Previously, we and others have extensively identified two mutually exclusive subpopulations of tumor-initiating cells purified only from the core of GBM tumors[12–15]. These populations of RICs presumably possess both tumor-initiating potential and preferential therapeutic resistance[16,17].

Multiple populations of GBM cells not only coexist within a single tumor but also cooperatively (or competitively) produce a variety of extracellular signals, elevating the complexity of the disease. The molecules secreted by one population may have a major impact on the cells from another population. Previously, we demonstrated intercellular crosstalk between apoptotic GBM cells located in the necrotic zone (tumor core) and the surrounding surviving cells via secreted exosomes. This intercellular signals potentiated tumor-initiating capacity in GBM through the switch of splicing isoforms in recipient cells[18] Other reports have also documented GBM cell crosstalk with endothelial cells[19], immune cells[20–22] etc. and GBM cells[23], collectively highlighting the impact of intercellular crosstalk on tumorigenicity and therapeutic resistance. However, the precise role of intratumoral crosstalk between spatially distinct tumor cells and its underlying mechanisms remain poorly understood.

Histone deacetylases (HDACs) are a class of molecules known to be associated with extracellular signals in tumor cells. The different isoforms of HDAC have been demonstrated to alter the protein content in exosomes[24] and the repertoire of the soluble proteins secreted by tumor cells[25,26]. In GBM, the intracellular function of HDACs is known to regulate the DNA damage response[27] and brain parenchyma invasion[28] by modulating the NF-κB and other pathways[29], Nonetheless, the effects of HDACs on intratumoral spatial crosstalk remain unknown.

In this study, we investigate the intracellular crosstalk between tumor core cells and the surgically inaccessible edge-located tumor-initiating cells. A set of large-scale small molecule inhibitor screens and RNA-sequencing data lead us to hypothesize that the presence of intercellular crosstalk mediated by HDAC1 provokes the aggressive transformation of the edge cells for tumor initiation.

## Results

**The edge and core GBM tissue exhibits distinct molecular properties.** Previously we identified two distinct tumor-initiating cell (TIC) subtypes, localized in the tumor edge and core[11]. In this study, we extended our prior work by specifically focusing on TICs related to the tumor edge, which presumably act as a major source of tumor recurrence (RICs). To isolate core and edge GBM tissues, the senior author (IN) performed an MRI-guided localized biopsy of GBM tissues from three newly diagnosed *IDH1*-wild-type GBM patients under an awake setting. As shown in Fig. 1a, the Gadolinium (Gd)-enhancing lesion of the GBM tumor represents the core and the T2-FLAIR highlights the tumor edge area hosting tumor-initiating cells. The edge and core tissue samples exhibited distinct histopathological characteristics: the edge tissues consisted of scattered infiltrating tumor cells embedded within largely normal brain parenchyma with few areas of reactive gliosis, whereas core tissues consisted of significant grayish necrotic regions, some hyper-vascular regions, and densely packed pseudopalisading and rapidly proliferating tumor cells, as showed by low-magnification images and H&E staining (Supplementary Fig. 1a, b). Immunohistochemical staining (IHC) demonstrated that edge samples are enriched with Olig2$^+$ cells, while core samples show higher expression of CD109—a putative core marker[11]. (Supplementary Fig. 1c). Next, we utilized RNA sequencing (RNA-seq) to transcriptionally profile matched GBM edge and core tissues, as well as non-tumor brain tissues derived from epilepsy patients. Principal component analysis (PCA) of RNA-seq data demonstrated a substantial gene signature difference between edge and core tissue (Fig. 1b). Sequencing data revealed uniquely elevated markers attributable to the GBM core (*CD44, MYC, HIF1α, VIM, ANXA1, CDK6,* and *JAG1*) and the GBM edge (*OLIG1, TC2, SRRM2, ERBB3, PHGDH,* and *RAP1GAP*) (Fig. 1c). Of note, both edge and core related marker sets included proneural (PN) and mesenchymal (MES) transcriptional subtype-related genes, indicating that edge tumor cells are not solely composed of PN cells and similarly that core cells are a mixture of multiple tumor subtypes, suggesting that the edge-core axis is clearly distinct from the PN-MES axis. Next, we referred to the IvyGAP Clinical and Genomic database[6], a collection of multiple regional samples of GBM tissues, to evaluate gene signatures of the tumor core, edge, and normal tissues (n = 9). Analysis demonstrated significantly higher cellular tumor (CT, n = 30) gene signature expression in our core tissues, whereas leading edge (LE, n = 19) signature expression was higher in our edge and normal brain tissues (Fig. 1d) On the other hand, the microvascular proliferation and pseudopalisading necrosis signatures did not show a significant difference (Supplementary Fig. 1d). Finally, we

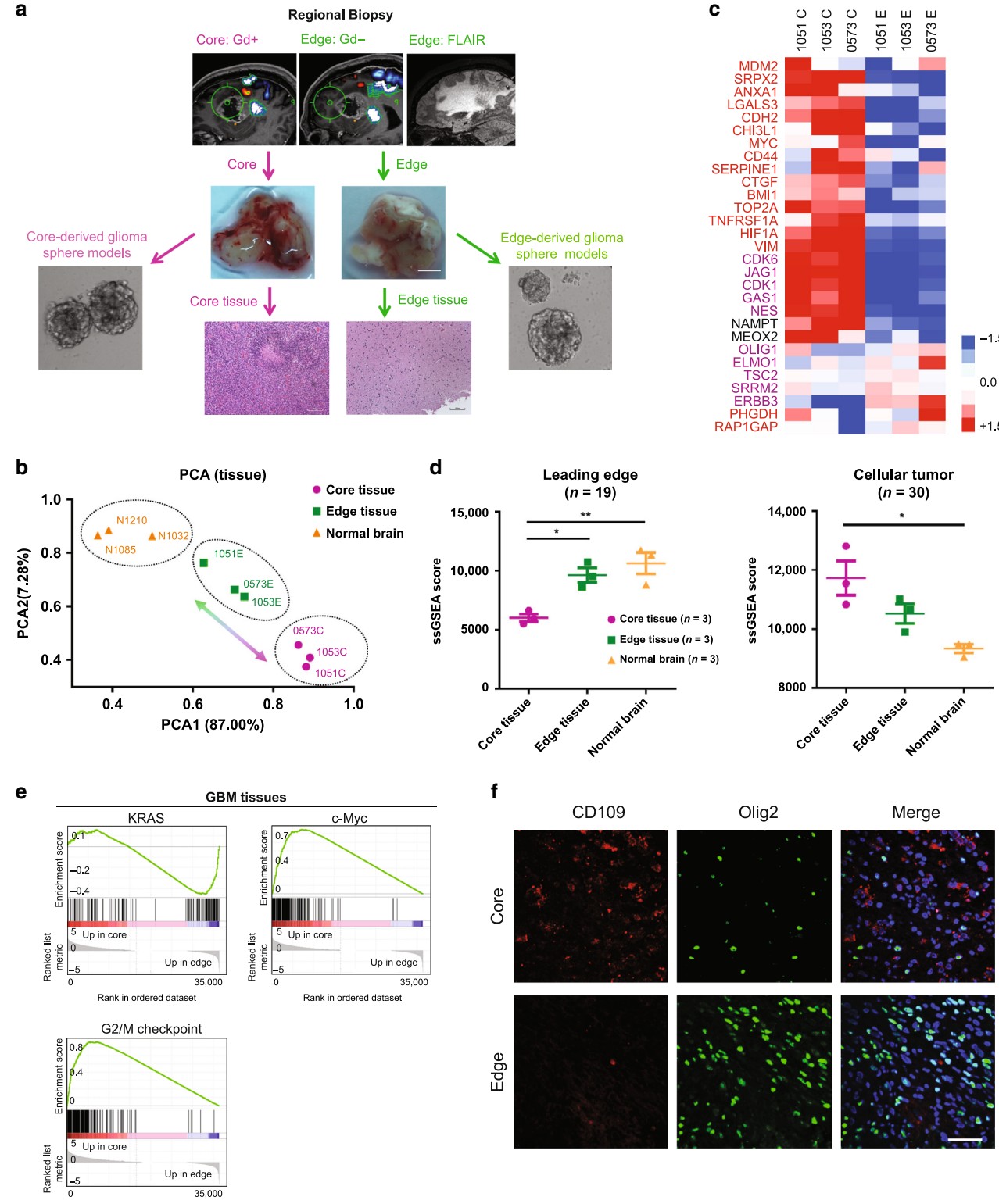

utilized gene-set enrichment analysis (GSEA) for assessment of key regulatory pathways. Activation of the KRAS pathway was evident in edge tissues, whereas activation of c-Myc and G2/M checkpoint pathways was observed in the core (Fig. 1e). Collectively, these analyses indicated that regionally specified tumor samples defining the edge-core axis demonstrated unique gene signatures distinct from the previously identified PN-MES axis[12–15].

To further compare edge/core signatures in association with the conventional transcriptomic subtypes, we tested the relationship between the transcriptional subtypes of GBM (PN, MES, or Classical (CL)) and the presence of proposed edge and core markers. We performed IHC staining of tissue microarrays from 61 GBM patients (PN = 31 patients, MES = 10 patients, CL = 11 patients, unclassified = 9 patients) with antibodies against Olig2 and CD109 (Supplementary Fig. 1e). Analysis showed that

**Fig. 1 GBM cells at the invading edge exhibit a molecular signature distinct from those localized in the core. a** Schematic representation of the experimental models established in the current study. Gadolinium (Gd)-enhanced T1-weighted and FLAIR MRI images of regional biopsy from edge and core regions of GBM (upper); representative surgical specimens from edge and core tumor tissues (middle), scale bar 2 mm; representative H&E staining of surgical specimens from edge and core tumor tissues (lower), scale bar 500 μm; representative images of neurosphere cultures (left and right). **b** Principal component analysis (PCA) of tumor tissue RNA-seq data of 3 paired GBM edge and core tissues and normal brain tissues derived from epilepsy surgery. **c** Heatmap of RNA-seq data demonstrating the differentially expressed genes. **d** Single-sample gene-set enrichment analysis (ssGSEA) of normal brain, GBM edge and GBM core tumor tissues using cellular tumor (CT) (left) and leading edge (LE) (right) gene signatures from Ivy Glioblastoma Atlas Project database. $n = 3$ independent samples per group; *$p < 0.05$, **$p < 0.01$ using one-way ANOVA followed by Tukey's post-test. Data are mean ± s.d. **e** Gene-set enrichment analysis (GSEA) of core tissues, compared to edge tissues. Gene sets shown include c-Myc, G2/M checkpoint and KRAS-associated genes. **f** Representative immunofluorescent staining of edge and core human GBM tissues for Olig2 (green), CD109 (red) and DNA (blue). Scale bar 50 μm.

more than half of the GBM samples (38 out of 61) demonstrated coexistence of presumed edge-like (Olig2+) and core-like (CD109+) cells (Supplementary Fig. 1f), indicating that edge and core signature may be relevant to all subtypes of GBM. Of note, MES tumors had significantly lower expression of Olig2 and higher expression of CD109 (Supplementary Fig. 1g, h), which is consistent with our previous report[11]. Finally, we used confocal microscopy to study the distribution of edge-like and core-like GBM cells with high-spatial resolution in different regions of the same tumor. As expected, tumor core was enriched for CD109+ cells, while tumor edge contained more Olig2+ cells (Fig. 1f). Importantly, only a few cells showed simultaneous staining for both markers, which may be due to unspecific staining rather than the simultaneous expression of Olig2 and CD109.

**Regionally specified GBM sphere models phenocopy intratumor spatial identities in vivo.** Our transcriptomic and immunohistochemical characterization of GBM edge and core cells showed a clear difference in molecular signature. This, however, raised the question of whether this difference might arise merely due to the variations in the proportion of normal and/or reactive non-tumorigenic cells in these tissue samples. Therefore, we used both edge and core tissue from GBM patients (1051 and 101027) to establish in vitro tumor sphere cultures. Then we utilized these short-term cultures for injection into immunocompromised mouse brains to test if they retain their spatial identity. To our surprise, IHC for human mitochondrial marker showed the presence of tumor cells in the injected site when we used core-derived 1051 sphere cells, whereas most of the edge-derived 1051 cells infiltrated into the corpus callosum—one of the major brain regions of GBM cell invasion[30] (Fig. 2a). Based on the GSEA data in Fig. 1, we then performed IHC with these models using antibodies against KRAS, c-Myc, and CHEK1. Consistent with the GSEA data, KRAS was preferentially expressed in the edge tumor cells and both c-Myc and CHEK1 were present in the core regions (Fig. 2b), To further validate these results, we developed another model by using slice cultures of neonatal mouse brains. When patient-matched core and edge cells (1051) were placed on top of these slice cultures, edge-derived cells rapidly moved toward the brain areas proximate to vasculature, while core-derived cells were randomly distributed in these slice cultures without any affinity toward vascular structure (Fig. 2c and Supplementary Movies 1 and 2). It is important to note that edge-derived cells did not necessarily move further away than the core counterparts in this model, suggesting that these phenotypic differences were not merely due to the more infiltrative nature of the edge cells. In fact, an in vitro migration assay demonstrated that the core-derived cells showed more migration capacity in vitro than the edge-derived cells (Supplementary Fig. 2a, b). Next, we labeled core spheres with a lentivirus vector encoding mCherry and co-injected these cells together with unlabeled edge cells into mouse brains. As expected, both IHC (Fig. 2d) and

immunofluorescence (Fig. 2e) for mCherry showed that core-derived tumor cells almost exclusively localized as a tumor mass in the injected site, whereas human mitochondrial staining revealed the presence of unlabeled human cells (edge-derived GBM cells) in broader portions of the mouse brains, even in the contralateral side through the corpus callosum. Taking together, these data indicate that edge/core signature is an intrinsic property of GBM cells, which persists after in vitro culture where tumor microenvironmental factors (such as extracellular signals from normal or apoptotic cells, hypoxia, etc.) are not present.

Experiments described above were performed on short-term in vitro cultures of regionally derived GBM spheres. To test if edge/core signature remains even after relatively long-term in vitro cultivation (>30 passages), we next analyzed non-matching GBM sphere lines that were established earlier from regionally undefined regions of tumors. Characteristics of all sphere lines used in the study are provided in Supplementary Data 1. First, we tested the expression levels of previously identified markers of edge (CD133 and Olig2[10,11]) and core (CD109[11]) in GBM spheres derived from 12 different patients (Supplementary Fig. 3a, b). Results of this experiment allowed us to subdivide these sphere lines into "core-like" (20, 1005, 1020, 267) and "edge-like" (157, 711, 1027, 1051, 1079) subgroups. Consistent with these data, the regionally specified edge spheres demonstrated higher CD133 expression and lower expression of CD109 and CD44, compared with the core counterparts (Supplementary Fig. 3c, d). Next, we performed RNA-seq of edge/core and edge-like/core-like spheres. PCA demonstrated significant similarity in gene expression of edge and core spheres with the edge-like and core-like counterparts (Fig. 2f). Of note, GSEA showed that c-Myc and G2M checkpoint were among the top differentially upregulated pathways in the core or core-like lines, while KRAS was identified in the edge or edge-like lines (Fig. 2g). As expected, co-injection of the GFP-labeled core-like spheres with mCherry-labeled edge-like spheres recapitulated the previous results with the matched core and edge sphere lines (Supplementary Fig. 3e). Collectively, these data indicate that spatial identity of the tumor core and edge cells is likely to be retained from the original tumor to sphere cultures and subsequent mouse xenografts (Supplementary Fig. 3f).

**Core-like GBM cells promotes malignancy of edge-like counterparts.** Advances in neurosurgical technologies (e.g., fluorescence-guided surgery) has enabled us to resect most of the enhancing core of GBM tumors. Nonetheless, certain regions of the brain remain inaccessible (i.e., thalamus). In fact, analysis of clinical data from 15 recent GBM cohorts (Supplementary Data 2) revealed that only one-third of GBM surgeries have achieved complete resection of the enhancing core lesion. More importantly, failure of complete resection of the enhancing lesions leads to substantially worse prognosis reducing the mean overall survival from 15.2–28.6 months with complete resection

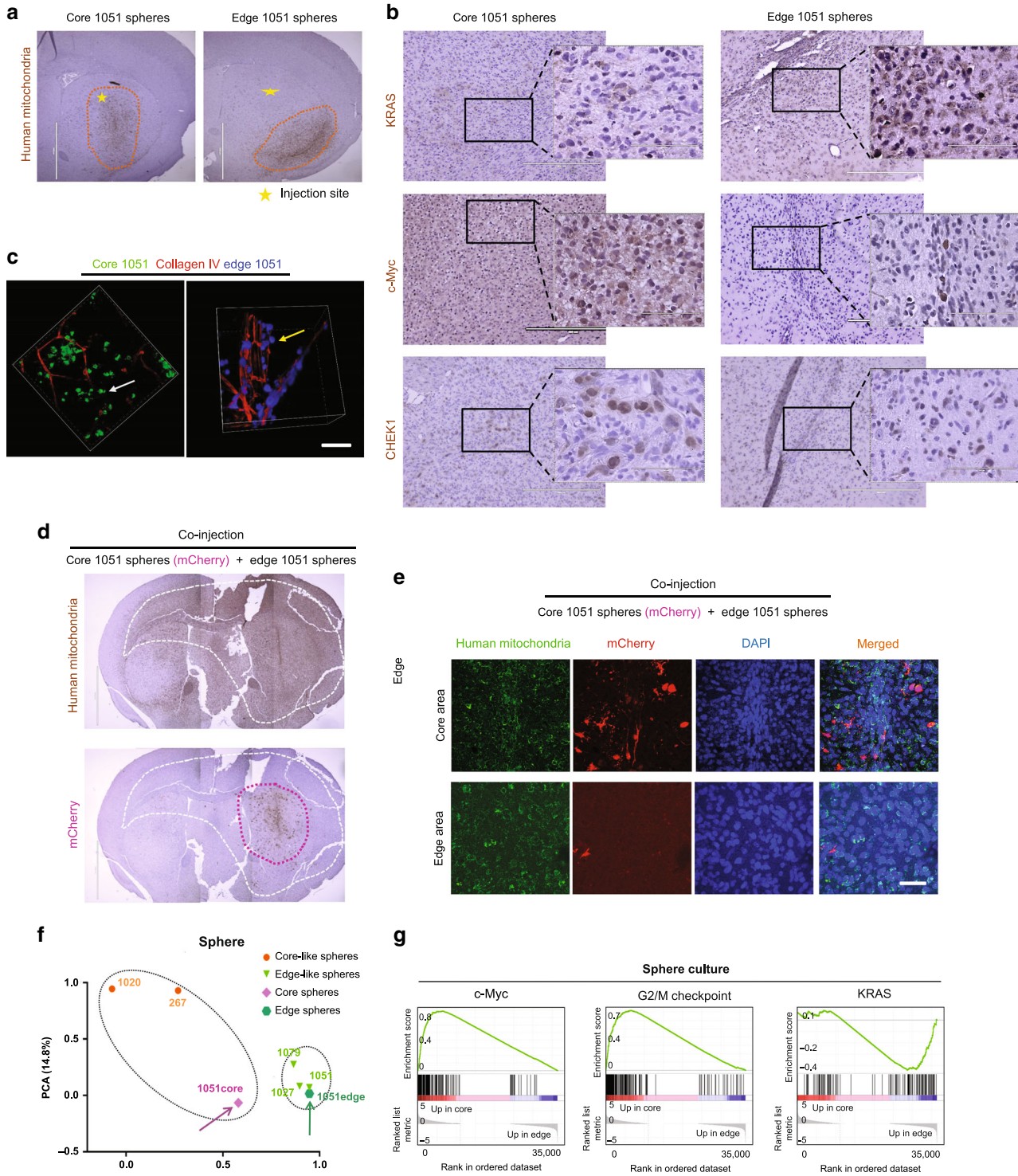

**Fig. 2 Regionally specified GBM cells phenocopy intratumoral spatial identities. a** IHC staining of mouse brains injected with edge or core 1051 GBM spheres for human mitochondria. Scale bar 2 mm. **b** IHC staining of mouse brains injected with edge or core 1051 GBM spheres for KRAS (upper), c-Myc (middle) and CHEK1 (lower). Scale bar 400um. **c** Representative immunofluorescence (IF) images of mouse brain slice culture seeded with edge 1051 (blue, yellow arrow) or core 1051 (green, white arrow) sphere cells and stained for Collagen IV to label blood vessels (red). Scale bar 50 μm. **d** IHC staining of a mouse brain co-injected with edge (unlabeled) and core (mCherry labeled) 1051 GBM spheres (ratio 1:1) for mCherry (lower, violet circle) and human mitochondria (upper, white circle). Scale bar 2 mm. **e** IF staining of the same samples as in "**d**" for human mitochondria (green), mCherry (red) and nucleus (blue). Scale bar 50 μm. **f** PCA of gene expression in edge, edge-like, core and core-like GBM sphere lines using set of 96-genes (32 each for proneural, mesenchymal and classical subtypes). **g** GSEA of core/core-like GBM spheres, compared to edge/edge-like GBM spheres. Gene sets shown include c-Myc, G2/M checkpoint, and KRAS-associated genes.

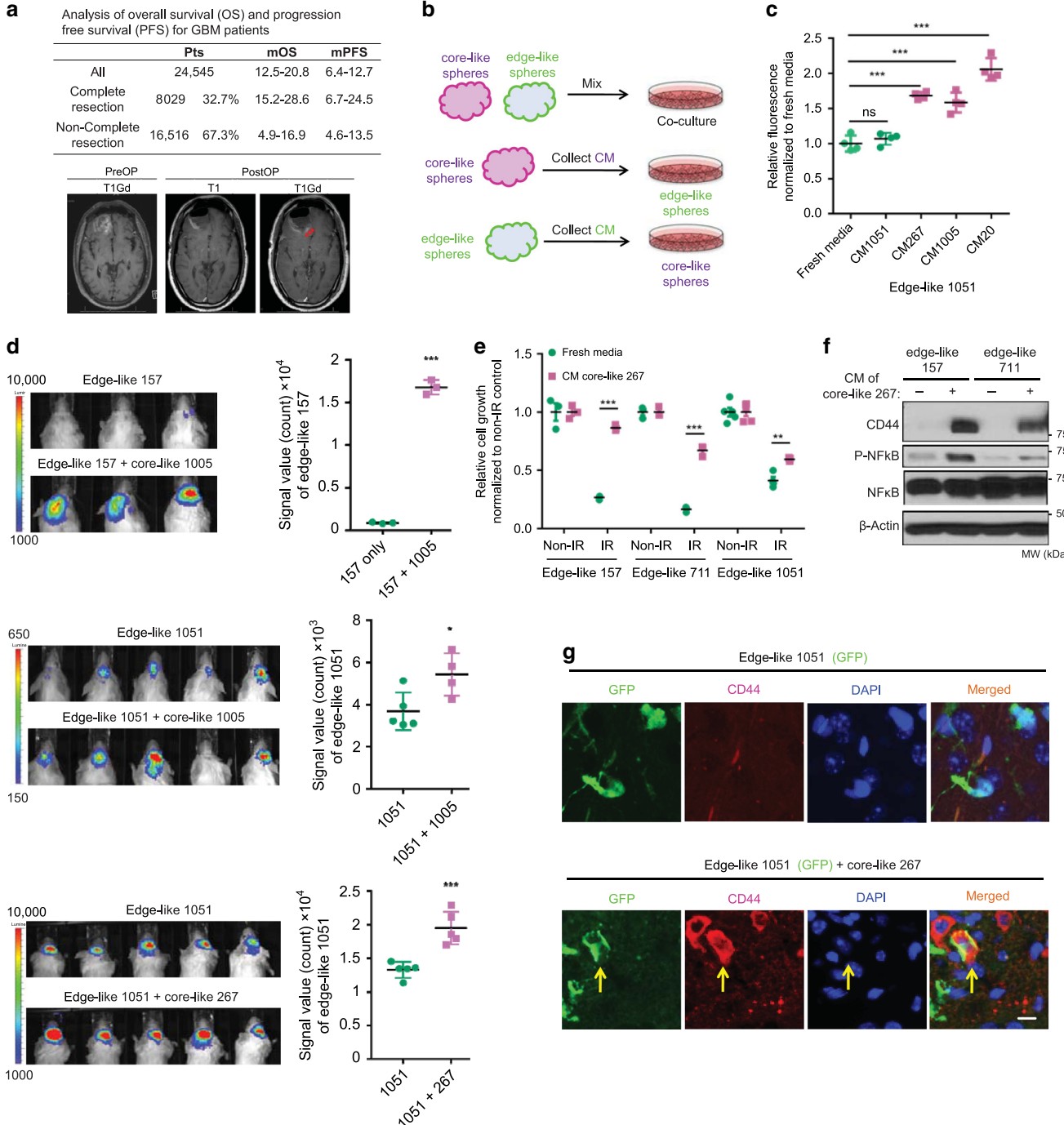

**Fig. 3 Intercellular signaling from core-like GBM cells provokes aggressiveness of their edge-like counterparts. a** Table comparing overall survival (OS) and progression free survival (PFS) of GBM patients that underwent complete or non-complete resection (upper). The representative MRI image of pre- and postoperative brain, demonstrating residual enhancing lesions after surgery (lower). **b** Schema of the in vitro experiments with conditioned medium (CM). **c** In vitro cell growth assay of edge-like 1051 GBM spheres treated with/without CM from edge-like 1051 or core-like 267/1005/20 GBM spheres. Data are mean ± s.d., $n = 4$ independent samples per group.; ns, not significant; **$p < 0.001$ using one-way ANOVA followed by Tukey's post-test. **d** Bioluminescence imaging (BLI) of mice intracranially injected with luciferase-labeled 1051 ($n = 5$ animals) or 157 ($n = 3$ animals) edge-like GBM spheres alone or together with unlabeled core-like GBM spheres (267, 1005) (ratio 95:5) (left). Quantification of BLI signal in mice (right). **e** In vitro cell viability assay of edge-like 157, 711, and 1051 GBM spheres pretreated with CM from core-like 267 GBM spheres and irradiated (IR) or left non-irradiated (non-IR). $n = 4$ independent samples per group. **f** Western blot (WB) for p65, phosphorylated p65 (p-p65) and CD44 using edge-like 157/711 GBM spheres treated with or without CM from core-like 267 GBM spheres. **g** IF staining for nucleus (blue), GFP (green) and CD44 (red) of mice brains bearing intracranial tumors developed from edge-like 1051 GBM spheres (labeled with GFP) alone, or co-injected with core-like 267 GBM spheres (unlabeled). Scale bar 20 μm. **d**–**e** Data are mean ± s.d. Significance was calculated by unpaired, two-tailed *t*-test with *$p < 0.05$; **$p < 0.01$; ***$p < 0.001$.

to 4.9–16.9 months with non-complete resection (Fig. 3a). Therefore, even a small residual tumor core may induce much worse outcomes for the patient. Such a significant decrease in survival is unlikely to be explained by the simple add-up effect of the remaining core tumor, rather it may indicate the presence of pro-tumorigenic trans-cellular crosstalk between residual core and adjoining infiltrating edge.

To test this hypothesis, we first utilized the edge-like and core-like cells to investigate whether communication between edge and core cells may contribute to tumor recurrence (Fig. 3b). Co-culture of edge-like 157 GBM spheres (mCherry labeled) with increasing proportions of core-like 267/28 GBM spheres demonstrated a growth-promoting effect of core-like cells on their edge-like counterparts (Supplementary Fig. 4a). To test whether a direct cell-cell contact is required for the observed effect, we cultured edge-like 1051 GBM spheres in the presence of conditioned media (CM) from core-like 267/1005/20 cells. In this experiment, CM was able to enhance the growth of edge-like spheres, indicating that a direct contact is not obligatory for the growth-promoting effect of core-like cells (Fig. 3c). In contrast, no noticeable change in proliferation was observed following application of CM from edge-like 157 GBM spheres to core-like 267 GBM spheres (Supplementary Fig. 4b). To further verify these data in vivo, we intracranially co-injected luciferase-labeled edge-like GBM spheres and unlabeled core-like spheres at a ratio of 95:5 in immunocompromised mice, mimicking the likely proportion of edge and core tumor cells following surgical resection. Bioluminescence imaging (BLI) revealed that the co-injection group demonstrated substantially greater tumor progression compared to mice injected with edge-like spheres alone (Fig. 3d). Consistent with these findings, staining for cell proliferation marker Ki67 was significantly higher in the co-injection group (Supplementary Fig. 4c).

To further characterize the effect of core-like GBM spheres on edge-like GBM spheres, we found that exposure to CM from core-like spheres increases radiation (IR) resistance of edge-like cells (Fig. 3e), enhances their motility (Supplementary Fig. 4d), and upregulates both RNA (Supplementary Fig. 4e) and protein levels of the core-associated marker CD44[11] with activation of nuclear factor-κB (NF-κB) (Fig. 3f). Consistent with the in vitro findings, elevation of the core-associated marker CD44 was observed in mouse GBM tumors by co-injection of edge-like GBM spheres with core-like spheres at a ratio of 95:5, but not edge-like alone (Fig. 3g and Supplementary Fig. 4f). Collectively, these data indicate that the intercellular signaling from core-like GBM cells promotes growth and IR resistance of edge-like GBM cells accompanied by the gain of core marker expression in vitro and in vivo.

**Regionally specified GBM core cells promote growth of edge cells.** Given our findings of a pro-tumorigenic effect of core-like on edge-like cells (Fig. 2), we examined whether similar growth-promoting effect is observed on the edge spheres. To test this, CM obtained from core spheres (patients 1051 and 101027) were added to their edge counterparts from the matched patients. Results from this experiment (Fig. 4a) demonstrated significant promotion of growth of edge spheres. To verify this finding in vivo, we intracranially co-injected edge and core GBM spheres at a ratio 9:1 in immunocompromised mice. Similar to the findings with co-injection of edge-like and core-like spheres, the presence of core GBM spheres significantly increased tumor growth (Fig. 4b).

From a therapeutic perspective, we asked whether the regionally specified GBM sphere lines exhibit attributable, clinically significant properties, such as distinct IR resistances. Utilizing in vitro cell viability assay of irradiated (4 Gy) core and edge GBM spheres, we found that core GBM spheres exhibit higher IR resistance than the edge spheres (Fig. 4c). This result was consistent with higher IR resistance of core-like spheres (Supplementary Fig. 4g). Similar to the gain of IR resistance by edge-like spheres (Fig. 3), edge spheres demonstrated elevated IR resistance following exposure to CM from core spheres (Fig. 4d). In addition, after the exposure to core CM, edge cells exhibited elevated expression of the core-associated marker CD44[11], consistent with our prior findings (Fig. 4e). To validate this data in vivo, we intracranially injected 1051 core or edge GBM spheres in mice. Following 2.5 Gy IR each day for 4 consecutive days, we compared BLI signals of the edge and core sphere-derived tumors. We observed higher BLI signals from the core sphere-derived tumors compared to their edge counterparts (Fig. 4f). Interestingly, when we co-injected luciferase-labeled 1051 edge with unlabeled 1051 core GBM spheres into mouse brain, we observed significantly elevated IR resistance of the co-injected edge cells compared to the edge sphere injection alone (Fig. 4g). Collectively, these findings demonstrate that both core and core-like spheres produce similar pro-tumorigenic intercellular signals affecting the edge GBM cells.

**HDAC1 participates in the pro-tumorigenic effect of core cells.** To elucidate the molecular mechanisms for the intercellular signaling from core to edge tumor cells, we established an in vitro co-culture mixed-sphere system of edge-like (labeled with green fluorescence protein (GFP)) and core-like (labeled with yellow fluorescence protein (YFP)) GBM cells. Then, we measured the ratio of these two fluorescence signals following exposure to 36,013 small molecules from four libraries (LOPAC Library (Sigma-Aldrich) (703 compounds), Drug Collection Library (3950 compounds), Naturally Isolated Compound Library (3200 compounds), Natural Crude Extract Library (28,160 compounds)). This screening identified 12 compounds with high proportion of GFP signal (edge-like) compared to YFP signal (core-like), suggesting those compounds suppress growth of core-like cells more potently than edge-like counterparts (Fig. 5a). Half of the identified compounds (6 out of 12) were inhibitors targeting class I (1, 2, 3 and 8) or class II (2a (4, 5, 7, and 9), 2b (6 and 10)) HDACs (Fig. 5a). To validate these data, we used second generation class I and II HDAC inhibitor, AR42[31,32], currently in clinical trial for acoustic schwannoma and meningioma (phase 0), as well as leukemia (phase I)[33], which has high blood–brain barrier permeability. First, we tested the effect of AR42 on sphere growth in vitro and in vivo. As expected, growth of core/core-like spheres was more efficiently suppressed by AR42 than their edge/edge-like counterparts in vitro (Fig. 5b and Supplementary Fig. 4h). In vivo, oral administration of AR42 significantly inhibited tumor growth of intracranially injected core-like 267 GBM spheres in immunocompromised mice (Fig. 5c). Additionally, AR42 demonstrated much higher inhibitory effect on core-like GBM cells compared with their edge-like counterparts (Fig. 5d and Supplementary Fig. 5a).

The shared molecular targets of HDAC inhibitors identified via this drug screening were HDAC1 and HDAC2, suggesting that one or both play roles in the growth of core-like tumor cells. To further explore this finding, HDAC1 and HDAC2 expression profiles were analyzed in our 59 matched longitudinal GBM sample collections (primary and recurrent cases). HDAC1 demonstrated a trend of elevated expression in CD109[up] recurrent patient tumors (Fig. 5e), accompanied by significantly elevated CCAAT/enhancer-binding protein beta (C/EBPβ) expression, which our recent study identified as a key

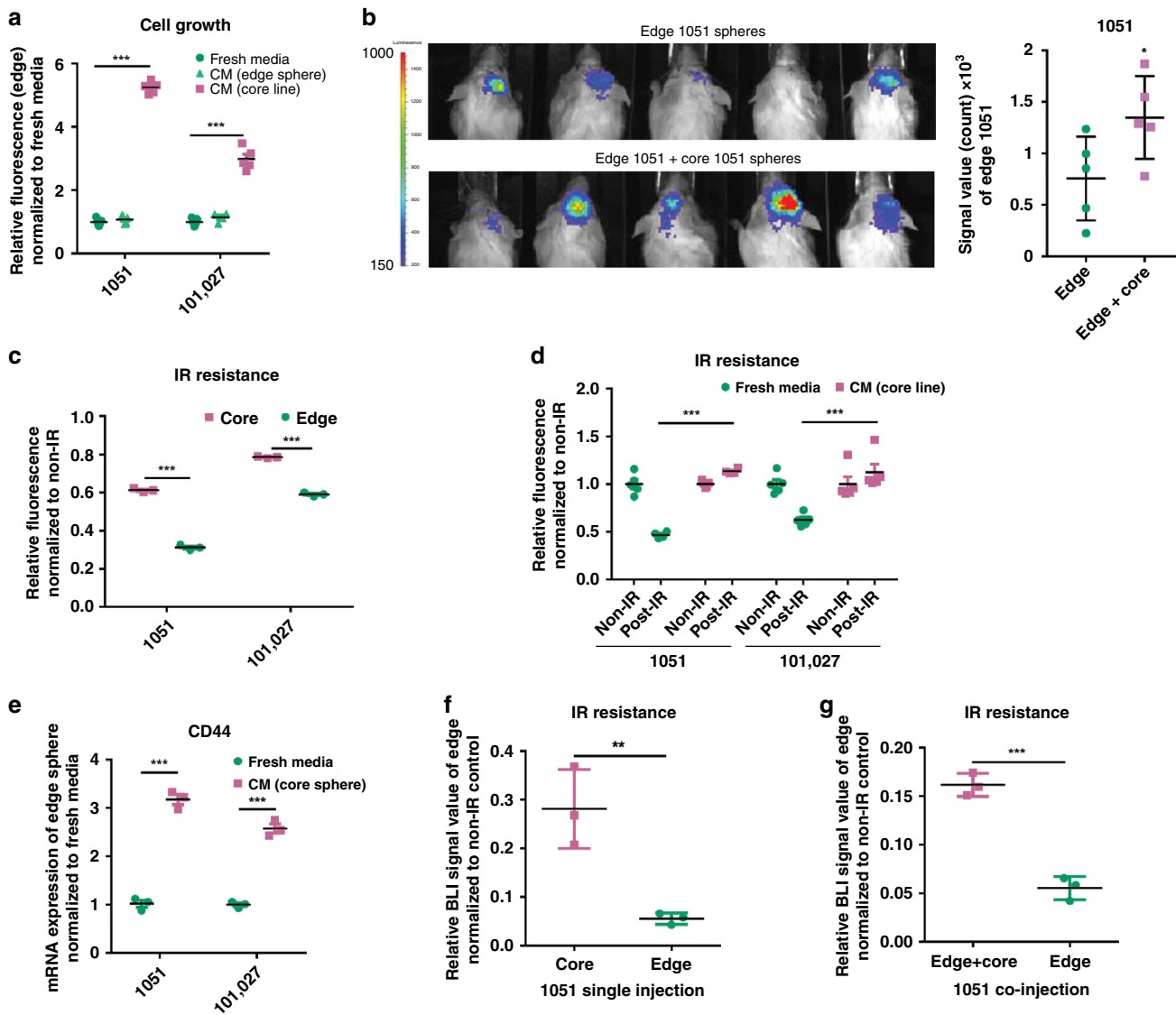

**Fig. 4 Regionally specified GBM cells demonstrate a similar growth promotion effect from GBM core to edge. a** In vitro cell growth assay of edge 1051 and edge 101027 GBM spheres treated with CM from core or edge spheres. $n = 5$ independent samples per group. **b** Bioluminescence imaging of mice intracranially co-injected with luciferase-labeled edge 1051 GBM spheres with or without unlabeled core 1051 GBM spheres (ratio 9:1) (left). Quantification of BLI signal in mice (right). $n = 5$ animals. **c** In vitro cell viability assay of irradiated (4 Gy; IR) or non-irradiated (non-IR) 1051 or 10127 edge and core GBM spheres. $n = 3$ independent samples per group. **d** In vitro cell viability assay of edge 1051/101027 spheres pretreated with CM from core spheres and subsequently irradiated. $n = 5$ independent samples per group. **e** qRT-PCR analysis of CD44 expression in edge 1051 and 101027 GBM spheres treated with fresh media or CM from core counterpart. $n = 3$ independent samples per group. **f** Quantification of BLI signal from mice intracranially injected with luciferase-labeled edge or core 1051 GBM spheres before and after irradiation. $n = 3$ animals. **g** Quantification of BLI signal from mice intracranially injected with luciferase-labeled 1051 cells edge alone or together with unlabeled core 1051 GBM spheres before and after irradiation. $n = 3$ animals. **a–g** Data are mean ± s.d. Significance was calculated by unpaired, two-tailed $t$-test with $*p < 0.05$; $**p < 0.01$; $***p < 0.001$.

transcriptional factor for CD109 expression[11] (Supplementary Fig. 5b). Furthermore, the RNA-seq data demonstrated that recurrent tumors with mesenchymal signature were associated with higher expression of HDAC1 (Supplementary Fig. 5c). This trend was also confirmed by IHC of 13 matched cases, although it was not statistically significant (Supplementary Fig. 5d). Furthermore, querying two publicly available databases, The Cancer Genome Atlas (TCGA) and The Repository for Molecular Brain Neoplasia Database (Rembrandt), revealed that HDAC1 expression was positively correlated with a higher grade of glioma (TCGA: $n = 213$ (Supplementary Fig. 5e); Rembrandt: $n = 446$ (Fig. 5f)). In addition, higher HDAC1 expression was consistently correlated with worse prognosis for GBM patients (Fig. 5g).

Conversely, HDAC2 expression was not significantly correlated with the grade of glioma (Fig. 5h) and was correlated with better prognosis (Fig. 5g). Consistent with these clinical data, edge-like GBM spheres exhibited higher messenger RNA (mRNA) expression of HDAC1 in comparison to core-like spheres (Supplementary Fig. 5f), validated at the protein level by western blot (Supplementary Fig. 5g). Finally, we performed immuno-fluorescent staining of human GBM tissue for HDAC1 and Olig2 or CD109. There was no colocalization between HDAC1 and Olig2, while staining for HDAC1 and CD109 showed similar pattern, indicating that HDAC1 is upregulated in core-like GBM cells (Supplementary Fig. 5h). Collectively, these data suggest that HDAC1 may contribute to the GBM core aggressiveness and raise

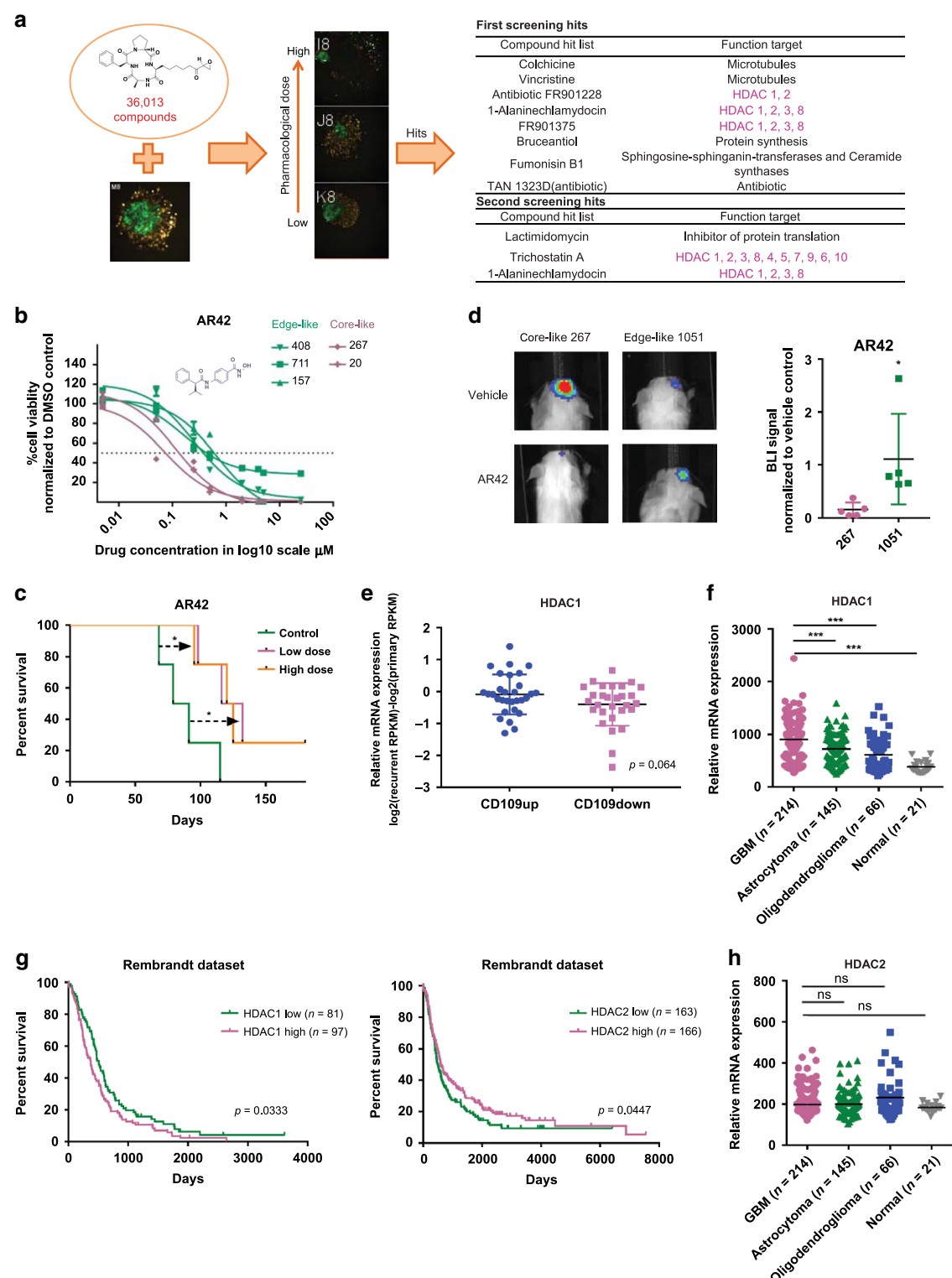

the possibility that HDAC1 provokes intercellular signals toward GBM edge cells.

**HDAC1 inhibition attenuates signaling from core GBM cells.**
As a proof-of-principle, we silenced the HDAC1 gene with short hairpin RNA (shRNA; Supplementary Fig. 6a) to investigate whether it plays a role in intercellular crosstalk from core to edge GBM cells. According to our data HDAC1 knockdown attenuated the ability of core-like CM to promote growth (Fig. 6a), IR

resistance (Fig. 6b), and CD44 expression (Supplementary Fig. 6b) in edge-like cells. Next, we utilized the HDAC inhibitor AR42 to check whether pharmacological inhibition can attenuate the growth-promoting effect. As expected, the presence of AR42 showed an effect similar to HDAC1 silencing. (Supplementary Fig. 6c). To validate these data in vivo, we intracranially co-injected luciferase-labeled edge-like 1051/157 GBM spheres with unlabeled core-like 267/1005 GBM spheres that were pre-infected with non-target shRNA (shNT) or shRNA against HDAC1 (shHDAC1) into the brains of immunocompromised

**Fig. 5 HDAC1 plays a role in the tumorigenic effect of core cells and is associated with poorer clinical outcome. a** Schematic representation of small molecule compound screening using in vitro co-culture mixed-sphere system; core-like and edge-like GBM spheres were labeled with YFP and GFP respectively (left). Hits from two independent screenings are presented in the table (right). Violet indicates HDAC inhibitors. **b** In vitro cell viability assay of edge-like 157/711/408 and core-like 267/20 GBM spheres treated with DMSO or AR42 at different concentrations. Data are mean ± s.d., $n = 4$ independent samples per group. **c** Kaplan–Meier survival curve of mice intracranially injected with core-like 267 GBM spheres followed by treatment with AR42 at different doses (10 mg/kg or 30 mg/kg). *$p < 0.05$, two sided log-rank test adjusted for multiple comparison; $n = 4$ animals. **d** Representative bioluminescence imaging of mice injected with edge-like 1051 or core-like 267 spheres and treated with vehicle or AR42 (left). Quantification of BLI signal in mice (right). *$p < 0.05$ using unpaired, two-tailed $t$-test. Data are mean ± s.d; $n = 5$ animals. **e** HDAC1 mRNA levels in matched longitudinal GBM samples (primary and recurrent tumors) grouped according to the CD109 expression level (up $n = 30$ patients or down $n = 29$ patients); $p = 0.064$ using unpaired, two-tailed $t$-test. Data are mean ± s.d;. **f** HDAC1 mRNA levels in glioma tumors ($n = 424$ patients) and normal brain samples ($n = 21$ patients) from Rembrandt datasets. ***$p < 0.001$ using one-way ANOVA followed by Tukey's post-test. The line is the median. **g** Kaplan–Meier survival curves of GBM patients subdivided by the level of HDAC1 expression ($p = 0.033$, two sided log-rank test; left) or HDAC2 expression ($p = 0.045$, two sided log-rank test; right). Data collected from the Rembrandt dataset ($n = 179$ patients). **h** HDAC2 mRNA levels in glioma tumors ($n = 424$ patients) and normal brain samples ($n = 21$ patients) from Rembrandt datasets. ns, not significant; using one-way ANOVA followed by Tukey's post-test. The line is the median.

mice at a ratio of 95:5. HDAC1 silencing in core-like spheres significantly diminished the growth-promoting effect on the edge-like counterparts in vivo (Fig. 6c, d). Immunofluorescent staining showed that co-injection of edge-like GFP⁺ 1051 with shNT-infected, but not with shHDAC1-infected, core-like 267 gave rise to GFP⁺/c-Myc⁺ tumor cells (Fig. 6e). To investigate the subsequent molecular outcome of silencing HDAC1, we performed RNA-seq of core-like GBM spheres infected with shNT or shHDAC1. GSEA of the RNA-seq data demonstrated that c-Myc and G2/M checkpoint pathways were downregulated by shHDAC1, similar to the results obtained with tissue RNA-seq (Fig. 1), suggesting c-Myc and G2/M as core-associated pathways (Fig. 6f). Other downregulated pathways included DNA replication, mismatch repair pathways, spliceosomes, and cell cycle progression (Supplementary Fig. 6d). The downregulated genes by HDAC1 silencing included those associated with core-ness, such as CD44 and CD109[11] and cell cycle-related genes such as MELK, FOXM1[34] and NEK2[35] (Fig. 6g). Collectively, these data indicate that HDAC1 plays a crucial role in the intercellular crosstalk between edge-like and core-like GBM cells and in the regulation of several cell survival/DNA repair pathways, including the c-Myc and G2/M checkpoint pathways, which are associated with the GBM core signature.

**Soluble CD109 mediates intercellular signals from core GBM cells.** To understand the nature of molecules that transmit signal from core to edge GBM cells, we pretreated CM with proteinase K (PK). CM pretreated with PK abrogated the growth-promoting effect (Fig. 7a) and morphological changes (Supplementary Fig. 7a) induced by the treatment of core-like CM to edge-like GBM spheres. These data indicate that signal from core to edge cells is most likely mediated by soluble proteins rather than lipids or extracellular vesicles. Thus, to identify the soluble factors secreted by core-like GBM cells whose expression is under the control of HDAC1, we first performed liquid chromatography-mass spectrometry (LC-MS) and identified 1309 proteins present in CM from either edge-like or core-like GBM spheres. The proteins identified from core-like spheres included soluble form of CD109 (sCD109), TGFBI[36], COL12A1 and SRPX[37] (Fig. 7b and Supplementary Data 3). Enzyme-linked immunosorbent assay (ELISA) of CM from regionally specified paired sphere lines from two different patients confirmed that core GBM cells secrete nearly 10 times higher amounts of sCD109 then their edge counterparts (Fig. 7c). Similar data were obtained for edge-like and core-like GBM spheres (Supplementary Fig. 7b). Importantly, molecular weight cut-off filtration of CM to remove extracellular vesicles did not significantly change the concentration of CD109 in CM, indicating CD109 indeed exist as a soluble form rather than a membrane-bound form (Supplementary Fig. 7c).

Next, we performed mass spectrometry analysis of CM collected from shNT- or shHDAC1-infected core-like spheres and demonstrated that CM from the HDAC1-silenced core-like spheres contains significantly less sCD109 comparing to the CM from shNT control (Fig. 7d and Supplementary Data 4). Previously, we identified CD109 (membrane-bound form) as a key marker for the core-localized tumorigenic cells[11]. The survival analysis of IVY GAP Clinical and Genomic Database dataset showed that the presence of CD109 in edge tissue correlates with significantly worse outcome for patients (Supplementary Fig. 7d). Therefore, among the all proteins identified in CM from core cells, we selected sCD109 for further investigation.

We confirmed that silencing of HDAC1 downregulates CD109 expression in core spheres, while overexpression of HDAC1 upregulates CD109 in edge spheres (Fig. 7e). Similar data were obtained using edge-like and core-like GBM sphere lines (Fig. 7f, g and Supplementary Fig. 7e). Importantly, we did not observe changes in CD109 expression upon knockdown of HDAC2 (Supplementary Fig. 7f). Clinically, TCGA dataset analysis confirmed the positive correlation between HDAC1 and CD109 expression, but not with HDAC2 expression ($n = 356$) (Supplementary Fig. 7g). Finally, we demonstrated that overexpression of HDAC1 significantly increases IR resistance of edge-like spheres (Fig. 7h).

We then sought to determine whether sCD109 promotes aggressiveness of edge tumor cells. Short-term exposure to recombinant sCD109 protein dramatically increased IR resistance of edge spheres isolated from two different patients (Fig. 7i). Similar data were obtained using edge-like 1079 spheres (Supplementary Fig. 7h). Consistent with this in vitro finding, intracranially co-injected edge-like 157 GBM spheres with core-like 1005 GBM spheres pre-infected with lentivirus encoding shRNA against CD109 (shCD109) showed diminished tumor growth compared to the edge-like 157 GBM spheres co-injected with shNT-infected core-like 1005 GBM spheres (Fig. 7j). Expression of c-Myc in GFP⁺ edge-like 157 cells was observed in the presence of co-injected core-like 1005 infected with shNT, but not with shCD109 (Fig. 7k). Collectively, these data indicate that sCD109 is secreted by core GBM cells and that this protein at least in part mediates the intercellular pro-tumorigenic effect from core GBM cells towards edge cells.

**Expression of CD109 is regulated by HDAC1-C/EBPβ complex.** Given that HDAC1 alters both mRNA and protein levels of CD109, we hypothesized that HDAC1 directly regulates the transcriptional activity of CD109 in GBM cells. Chromatin immunoprecipitation (ChIP) in core-like spheres detected an occupancy of HDAC1 at the CD109 promoter region (Fig. 8a). Given that our recent study identified C/EBPβ as a key

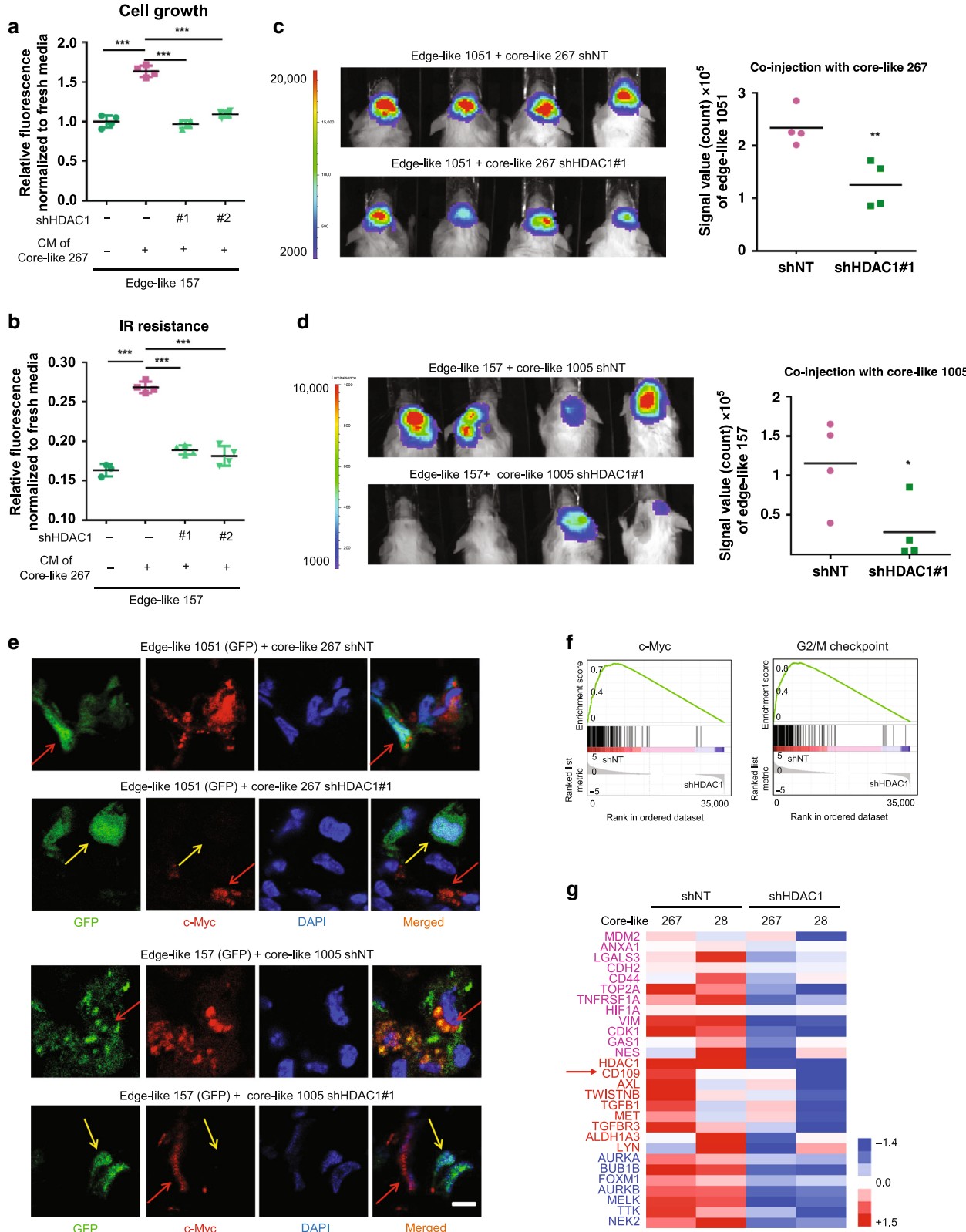

transcriptional factor for *CD109* expression[11] and that analysis of our matched longitudinal GBM samples indicated that C/EBPβ was significantly higher in CD109[up] recurrence group (Supplementary Fig. 5b), we examined whether occupancy of C/EBPβ at the promoter region of *CD109* is under the control of HDAC1 in core-like spheres. ChIP-PCR showed that shHDAC1 diminished the binding of C/EBPβ in two of three *CD109* promoter regions in

core-like 1005 (Fig. 8b) and core-like 267 (Supplementary Fig. 8a). We confirmed these data using regionally confined 1051 GBM spheres and demonstrated much higher enrichment of HDAC1 at *CD109* promoter site in core spheres, compared to the edge counterparts (Fig. 8c). To further validate the HDAC1 and C/EBPβ-combined regulation of *CD109* transcription, we confirmed protein complex formation between HDAC1 and C/EBPβ

**Fig. 6 Inhibition of HDAC1 attenuates the intercellular signaling from core to edge GBM cells. a** In vitro cell growth assay of edge-like 157 GBM spheres treated with CM from core-like 267 GBM spheres, which were infected with lentiviruses encoding shNT or shHDAC1. $n = 4$ independent samples per group. Data are mean ± s.d., ***$p < 0.001$ using one-way ANOVA followed by Tukey's post-test. **b** In vitro cell viability assay of edge-like 157 GBM spheres pretreated as in "**a**" and subsequently irradiated. $n = 4$ independent samples per group. Data are mean ± s.d., ***$p < 0.001$ using one-way ANOVA followed by Tukey's post-test. **c** Bioluminescence imaging of mice intracranially co-injected with luciferase-labeled edge-like 1051 GBM spheres and unlabeled shNT-infected or shHDAC1-infected core-like 267 GBM spheres (95:5 ratio) (left). Quantification of BLI signal in mice (right). $n = 4$ animals. **d** Same as in "**c**" for co-injection of luciferase-labeled edge-like 157 GBM spheres and unlabeled core-like 1005 GBM spheres, $n = 4$ animals. **e** IF staining for GFP (green), c-Myc (red), and nucleus (blue) of mice brains bearing intracranial tumors developed from co-injection of GFP-labeled edge-like 1051/157 GBM spheres and shNT-infected or shHDAC1-infected core-like 267/1005 GBM spheres. Scale bar 10 μm. **f** GSEA of shNT-infected core-like 267/28 GBM spheres, as compared to shHDAC1-infected spheres. Gene sets shown are c-Myc and G2/M checkpoint-associated genes. **g** Heatmap of RNA-seq data comparing expression of selected genes in core-like 267/28 GBM spheres infected with shNT or shHDAC1. CD109 indicated by red arrow. **c, d** Data are mean ± s.d. Significance was calculated by unpaired, two-tailed $t$-test with *$p < 0.05$; **$p < 0.01$.

---

in core-like 267 (Fig. 8d) and core-like 1005 GBM spheres (Supplementary Fig. 8b). This finding agreed well with previously published data that describes role of HDAC1-C/EBPβ complex in the regulation of transcription[38]. Finally, re-ChIP demonstrated that the protein complex of HDAC1 and C/EBPβ binds to the promoter region of the *CD109* gene (Fig. 8e). Collectively, these results indicate that HDAC1 positively regulates *CD109* in a C/EBPβ-dependent manner in core GBM cells.

## Discussion

By consolidating our focus on the transcriptional and phenotypic identities of GBM intratumoral spatial heterogeneity, this study developed a set of novel in vivo and in vitro models representative for the infiltrating edge—a difficult lesion to access unless we perform an awake surgery, as well as the surgically resectable contrast-enhancing core. We demonstrated that intercellular crosstalk from core (and core-like) GBM cells promotes aggressiveness of the edge counterparts in both matched and unmatched combinations. As one molecular mechanism, small molecule screening and downstream experiments identified HDAC1 as the initiator of the crosstalk and sCD109 as the acting mediator via HDAC1-C/EBPβ regulation.

Transcriptional subtyping in GBM has neither attributed a worse prognosis to any particular subtype, nor identified targetable specific gene mutations[37], hence, achievement of the practice-changing discoveries through this effort has not been made thus far. As a result, the presence of both intra- and intertumor transcriptomic heterogeneity has solely increased the scope of complexity of GBM[39]. That is one of the reasons why our research team has recently shifted our focus on the tumor edge for identification of targetable molecules for prevention of tumor recurrence following the inevitable incomplete resection of GBM cells via surgery[40–42].

However, our current understanding of the evolutionary development of GBM's spatial heterogeneity remains mostly, if not exclusively, constrained to the resectable, contrast-enhancing core region[5,42]. Utilizing MRI neuronavigation-based regional biopsies, we were able to collect regionally distinct GBM patient tissue samples without harming vital structures by an awake surgery, followed by establishment of regionally specified GBM (neuro)sphere culture models with derivative mouse tumor xenografts. Our analyses have suggested that these models carried over similar, if not identical, molecular signatures to those in the original tumor tissue. Our data for these spatially distinct gene signatures were supported by the information collected from two datasets: the IVY Glioblastoma Atlas[6] (42 patients) for regional characterization by laser capture gene expression analysis, and the Darmanis dataset[43] (4 patients) for single-cell RNA-sequencing from patient tumor tissue. Our edge-associated cells (tissue, sphere cultures, and derivative mouse xenografts) showed similarity to the PN GBM signature only to some degree, yet clearly

harbored a difference. On the other hand, our core-associated cells were a mixture of cells with all three subtypes, including MES signature. However, as previously established, clinical relevance of the PN-to-MES axis remains indeterminate at best. In contrast, there is no question that the edge-located tumor cells are the majority of, if not the only, tumor cells that subsequently develop lethal recurrence, which is a direct indication of the clinical significance of further characterizing this poorly studied cellular property.

Clinically, we practitioners almost always see that in patients' brains, uncountable concerning tumor edge lesions end up arising new core lesions over time as tumor recurrence causing patient mortality regardless of the clinical background and therapies. In the experimental setting, this edge-to-core transition appears to be partially, but not completely, accompanied by the post surgical PN-MES transition in tumor-initiating cells, which we have previously identified in GBM[13]. Given this observation, we find one unsolved question related to these two sets of axis (PN-to-MES and Edge-to-Core). Owing to the limitation in a reliable experimental recurrence model elucidating the evolutionary processes undertaken throughout edge GBM cells for core reformation, we cannot fully capture the array of pro-tumorigenic signaling events for tumor re-establishment in the recurrence development related or unrelated to PN-to-MES transition. Our data indicated that there is persistence of the spatial and phenotypic properties of GBM cells derived from the tumor edge and core lesions in alignment with that of the originating tumor's spatial identity. These findings raised the possibility that cell-intrinsic factors wrest control of spatial identity at an early time-point of tumor development, leading to the appearance of stable cell autonomous differences between core- and edge-located GBM cells. Therefore, once a GBM cell acquires edge or core identity, it can possibly be maintained even in the absence of tumor microenvironmental factors. At the same time, in agreement with our previous observation[18], this cell autonomous phenotype can also be affected by various paracrine signals from another population of GBM cells in ecosystem. We propose that upon tumor growth, various factors such as low pH and hypoxia can trigger the acquisition of the core phenotype due to phenotypic changes in various unknown cell types in ecosystem, which is then imprinted as cell-intrinsic mechanisms. Importantly, this study identified that these core cells can also disseminate some of their malignant properties to less aggressive edge GBM cells by passing a number of extracellular signals to support their ecosystem.

However, methods for the assessment of selective pressures resulting in end-point spatial differentiation of GBM cells at a pre-diagnostic time-point due to microenvironmental factors are limited. Hence, the factoring of microenvironmental conditions within the brain, predating surgical resection, is restricted in our in vitro and in vivo models. Another point is that potential gene

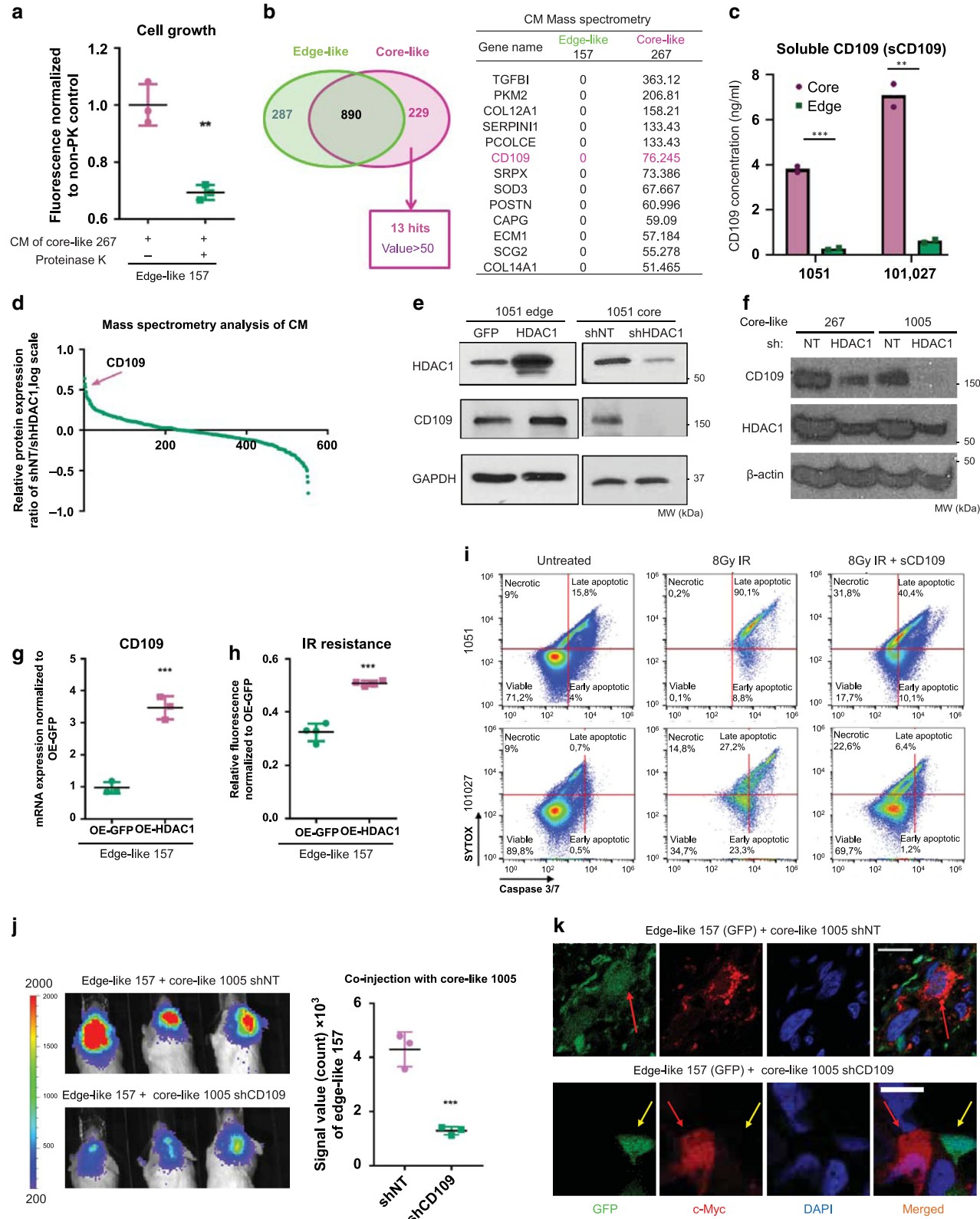

mutation-based alteration of spatial identity throughout the evolutionary process, as well as throughout growth of our models, was not evaluated in this study, thus the importance of specific genes and transcriptional regulators remains an open question. Additionally, given that surgical resection evokes a characteristic peritumoral immune response in the tumor microenvironment[44], it will be an important area of future investigation to explore the interaction of edge tumor cells and post surgical infiltration of immune cells. Nonetheless, via utilization of these models, the identification of cell-intrinsic and microenvironmental regulators determining spatial identity should be possible and represents a critical step for development of edge-targeting therapy in GBM, with potential therapeutic benefit in other spatially heterogenous cancers.

**Fig. 7 Soluble CD109 is a mediator of HDAC1-derived intercellular signals from core to edge GBM cells. a** In vitro cell growth assay of edge-like 157 GBM spheres treated with CM from core-like 267 GBM spheres incubated with or without proteinase K. $n = 3$ independent samples per group. **b** Venn diagram illustrating proteins identified by LC-MS/MS in CM from core-like and edge-like GBM spheres (left). Table showing the top 13 proteins exclusively detected in core-like CM (right). **c** Enzyme-linked immunosorbent assay (ELISA) for soluble CD109 in CM from 1051 and 101027 edge or core patient-derived GBM spheres. $n = 2$ independent experiments. **d** Graph demonstrating differences in the relative amount of proteins detected by LC-MS/MS in CM from core-like cells infected with shNT or with shHDAC1 encoding lentiviruses. **e** WB for HDAC1 and CD109 using 1051 core GBM cells infected with shNT or shHDAC1 lentiviruses or using 1051 edge GBM cells infected with GFP or HDAC1 encoding lentiviruses. **f** WB for CD109 and HDAC1 using core-like 267/1005 GBM spheres infected with shNT or shHDAC1 lentiviruses. **g** qRT-PCR analysis of CD109 expression in edge-like 157 GBM spheres infected with lentiviruses encoding GFP or HDAC1. $n = 3$ independent samples per group. **h** In vitro cell viability assay of edge-like 157 GBM spheres infected with lentiviruses encoding GFP or HDAC1 and subsequently irradiated (IR). $n = 4$ independent samples per group. **i** Flow cytometry analysis of caspase-3/7 activity and SYTOX staining in 1051 and 101027 edge or core spheres that were cultivated in a presence or absence of recombinant sCD109 for 3 days and subsequently irradiated with 8 Gy. **j** Bioluminescence imaging of mice intracranially co-injected with luciferase-labeled edge-like 157 GBM spheres and unlabeled shNT or shCD109-infected core-like 1005 GBM spheres (95:5 ratio) (left). Quantification of BLI signal in mice (right). $n = 3$ animals. **k** IF staining for GFP (green), c-Myc (red) and nucleus (blue) of mouse brains co-injected with GFP-labeled edge-like 157 GBM spheres and unlabeled shNT or shCD109-infected core-like 1005 GBM spheres. Scale bar 20 μm. **a–j** Data are mean ± s.d. Significance was calculated by unpaired, two-tailed *t*-test with **$p < 0.01$; ***$p < 0.001$.

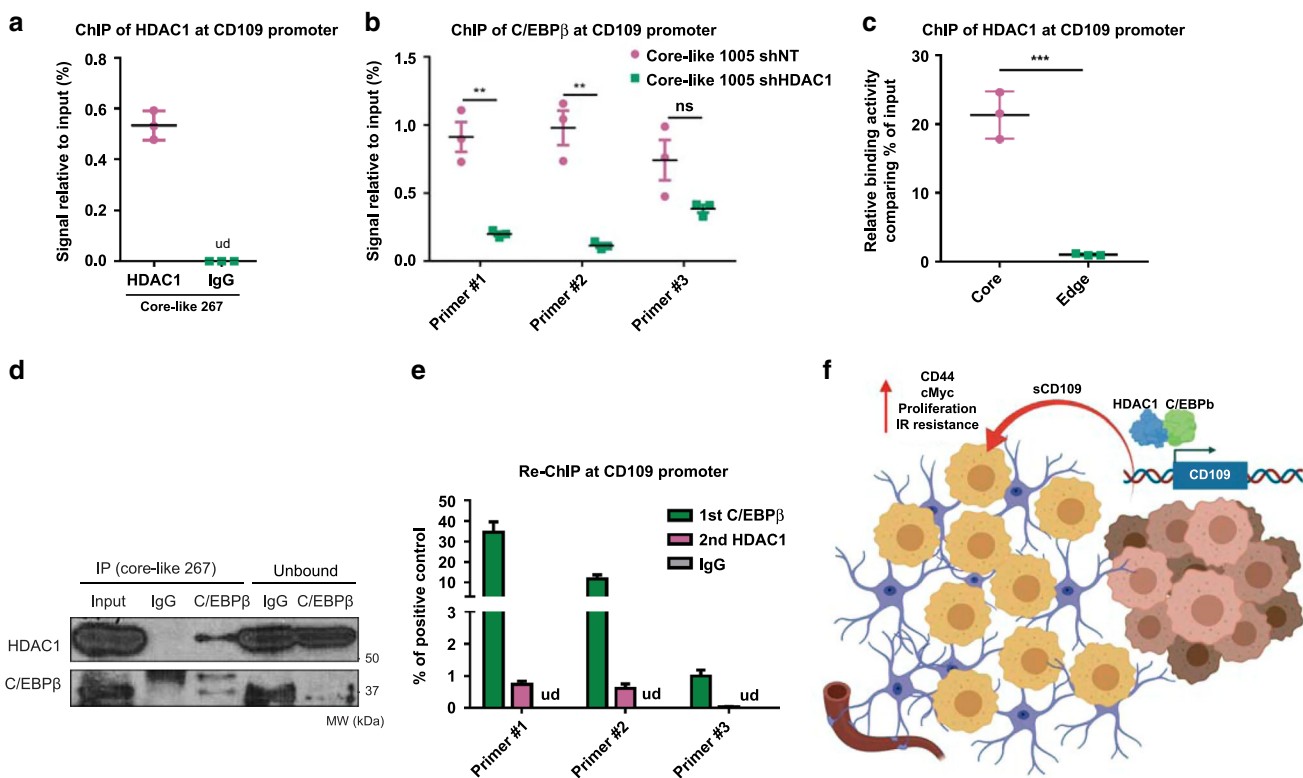

**Fig. 8 HDAC1 regulates transcription of *CD109* via C/EBPβ. a** ChIP analysis showing enrichment of HDAC1 at *CD109* promoter region in core-like 267 GBM spheres. ud- undetected, $n = 3$ independent samples per group. **b** ChIP analysis showing enrichment of C/EBPβ at *CD109* promoter region in core-like 1005 GBM spheres infected with shNT or shHDAC1, $n = 3$ independent samples per group; ns, not significant. **c** ChIP analysis showing binding of HDAC1 to *CD109* promoter region in 1051 core and edge GBM spheres. $n = 3$ independent samples per group. **d** Co-immunoprecipitation of HDAC1 in core-like 267 GBM spheres with antibodies against C/EBPβ. **e** Re-ChIP analysis using anti-C/EBPβ antibodies for first precipitation, followed by second immunoprecipitation with antibodies against HDAC1 and subsequent PCR with primers spanning promoter region of *CD109*. $n = 3$ independent samples per group. ud- undetected. **f** Proposed molecular mechanism of intercellular crosstalk between core and edge GBM cells via soluble CD109 protein. HDAC1-C/EBPβ-CD109 signaling induces upregulation of CD44 and c-Myc in edge cells and ultimately leads to the increased proliferation and therapy resistance. **a–e** Data are mean ± s.d. Significance was calculated by unpaired, two-tailed *t*-test not adjusted for multiple comparison with **$p < 0.01$; ***$p < 0.001$.

HDAC1 has been widely reported for its gene silencing role in multiple carcinomas[45,46]. Despite HDAC1 being recognized as a co-factor of HDAC2 for transcriptional silencing[47], studies have shown that their activation may also lead to distinct effects[46,48,49], evidenced here by the observed mutual expression profile of HDAC1 and HDAC2 in GBM. This study determined that HDAC1, but not HDAC2, positively regulates the transcriptional activity of *CD109*. Interestingly, this is contrary to its known gene

silencing role, but remains unexplained in this study and elucidating the mechanism underlying this effect will be an important point in future research. On the other hand, C/EBPβ, a contributor to the core signature via positive regulation of *CD109*[11], is reportedly deacetylated by HDAC1, increasing its binding ability[50]. IP and re-ChIP experiments confirmed that HDAC1 and C/EBPβ form a protein complex so that they occupy the promoter region of the *CD109* gene. Nonetheless, unanswered

questions remain. It was previously shown that HDAC inhibition affects the expression of a substantial number of genes in the human genome and according to our data, the level of the co-occupancy of HDAC1 and C/EBPβ on the *CD109* promoter detected by the ChIP experiment was rather low. Therefore, it is possible that HDAC1 may not be the main regulator of *CD109* in core GBM cells. Rather, it may mediate the effect of the more specific regulator of *CD109* expression that still has to be determined. In addition, the roles of *CD109* in the soluble and in the membrane-bound forms may be distinct in a context-dependent manner in GBM cells.

Clinically, our results indicated that specific inhibition of HDAC1 is a potential strategy for future combination treatment of GBM after surgical resection. There are several HDAC inhibitors in clinical trials such as vorinostat, trichostatin A or panobinostat, targeting class I, II, and IV HDACs. In our study, we used AR42 (class I and class II HDAC inhibitor) and shRNA specifically targeting HDAC1. Both were able to significantly decrease GBM growth both in vitro and in vivo. Thus, further development of HDAC1 inhibitors may contribute to future clinical treatment. These new drugs may prevent acquisition of the aggressive and highly resistant core phenotype and, therefore, improve the efficacy of conventional chemo- and radiotherapy.

In conclusion, this study demonstrated the presence of intercellular signals that affects tumor-initiating cells at the edge. These signals were provoked by neighboring residual core cells and promoted growth and radioresistance of the edge counterparts in a HDAC-CD109 dependent manner. Surgically inaccessible tumor edge retains hidden devils that kill patients. Investigation of resectable tumor cells alone would not uncover new cellular and molecular targets. Given this notion in mind, we would and should, make our progress toward identifying practice-changing therapeutic modalities targeting the edge-located tumor-initiating cells for patients with GBM.

## Methods

**Ethics**. This study was conducted under the approved Institutional Review Board (IRB) and Institutional Animal Care and Use Committee (IACUC) protocols in University of Alabama at Birmingham (UAB), University of California Los Angeles (UCLA) and MD Anderson Cancer Center (MDA). The IRB Protocol number at UAB was #N151013001. Clinical glioma specimens and normal brain tissue samples were collected in the Department of Pathology and Laboratory Medicine at UAB (sample 1005, 1014, 1020, 1027, 1037, 1051, 1079), UCLA (sample 157, 336, 339, 374, 408), MDA (sample 267, 20, 28, 711), and processed at the research laboratories after de-identification of the samples.

**GBM-derived neurospheres and cell culture**. Freshly resected GBM samples were dissociated and neurosphere cultures were established and cultivated in DMEM/F12 medium containing 2% B27 supplement, 1% Penicillin-Streptomycin solution, 2.5 μg/ml heparin, 20 ng/ml basic fibroblast growth factor (bFGF), and 20 ng/ml epidermal growth factor (EGF). bFGF and EGF were added twice a week and the cultural medium was changed every 7 days. The data with neurospheres were obtained with cells cultivated for no longer than 40 passages. The cell lines were tested negative for mycoplasma contamination.

**RNA isolation and quantitative real-time PCR**. mRNA was isolated by Trizol (Thermo scientific) according to the manufacturer's protocol. RNA concentration was determined using a Nanodrop 2000 Spectrophotometer (Thermo scientific). Complementary DNA (cDNA) was synthesized by using iScript reverse transcription supermix (Bio-Rad) according to the manufacturer's protocol. qPCR was performed on StepOnePlus thermal cycler (Thermo scientific) with SYBR Select Master Mix (Thermo scientific). Cycling conditions were 95 °C for 5 min, and then 40 cycles of 95 °C for 30 s, 60 °C for 30 s and 72 °C 30 s. Primer specificity was confirmed by visualizing DNA on an agarose gel following PCR. GAPDH was used as an internal control. Primer sequences are listed in Supplementary Data 5.

**Flow cytometry**. For the apoptosis assay cells were stained with CellEvent Caspase-3/7 Green Flow Cytometry Assay Kit (Thermo scientific) according to the manufacturer's protocol. After staining samples were analyzed by Attune NxT Flow Cytometer (Thermo scientific) with Attune NxT version 4.2 software. Data were processed with FlowJo version 10 software. The samples were gated by FSC-H and SSC-H to distinguish cells from debris (Supplementary Fig. 9a).

**Western blot**. Cells were lysed for 30 min on ice in RIPA buffer (Sigma) containing 1% protease and 1% phosphatase inhibitor cocktail (Sigma). Lysates were precleaned by centrifugation at $16,000 \times g$, 15 min, 4 °C. Protein concentration was determined using the Bradford method. Equal amounts of protein lysates (10 μg/lane) were fractionated by NuPAGE Novex 4–12% Bis-Tris Protein gel (Thermo scientific) and transferred to a PVDF membrane (Thermo scientific). Subsequently, the membrane was blocked with 5% Blotting Grade Blocker Non Fat Dry Milk (Bio-Rad) for 1 h, incubated with corresponding primary antibody overnight and then incubated with peroxidase conjugated secondary antibodies (GE Healthcare) for 1 h. Staining was visualized with Amersham ECL Western Blot System (GE Healthcare). Unprocessed films for each western blot are provided in Supplementary Fig. 9b.

**Immunocytofluorescence**. Cells were fixed with 4% PFA, permeabilized with 0.2% Triton-X, blocked with serum-free protein block solution (Dako) and incubated with corresponding primary antibodies for 1 hour at room temperature. Next, cells were incubated with Alexa Flour-conjugated secondary antibody for 1 h at room temperature and mounted in Vectashield mounting medium containing DAPI (Vector Laboratories). Images were captured on Nikon A1 confocal microscope using NIS-Elements version 4.00 software.

**Immunohistochemistry**. Tumors embedded in paraffin blocks were deparaffinized, and hydrated through an ethanol series. After microwave antigen retrieval in DakoCytomation target retrieval solution pH 6 (Dako), slides were incubated in 0.3% hydrogen peroxide solution in methanol for 15 min at room temperature to inhibit internal peroxidase activity. Next samples were blocked with serum-free protein block solution (Dako) and incubated with corresponding primary antibodies overnight at 4 °C. The next day slides were stained with EnVision + System—HRP labeled Polymer (Dako) and visualized with DAB peroxidase substrate kit (Vector Laboratories). Signals were detected using Olympus DP71 microscope.

**Cell viability assay**. Viability of GBM cells was determined using AlamarBlue reagent (Thermo scientific). Cells were seeded at 3000 cells per well in a 96-well plate, after indicated period of time AlamarBlue reagent was added into each well and 6 h later fluorescence was measured (Excitation 515–565 nm, Emission 570–610 nm) using a Synergy HTX multi-mode reader (BioTek).

**Wound healing assay**. Wound healing (scratch) assay was performed according to the standard protocol[18]. Briefly, glioma spheres were dissociated into single cells and plated on a laminin coated 6-well plate at $1.5 \times 10^6$ cells per well. The following day, cell cultures were scratched using a pipette tip. Forty-eight hours later, cells were visualized by light microscopy. These assays were performed three times.

**Brain slice culture**. The animals (postnatal P7) were rapidly sacrificed, the head briefly placed in 70% ethanol and the brains dissected. Under aseptic conditions, 350 μm-thick whole-brain (sagittal or coronal) sections are cut and collected in sterile medium. One-thousand fifty-one edge sphere cells (stained with PKH67 Green Fluo (Sigma) were seeded on the brain slices ($10^5$ cells) along with 5 μl matrix gel. Brain slices were incubated at 37 °C and 5% $CO_2$ for 2 days. Next, the slices were fixed for 8 h at 4 °C in 4% PFA and stained for collagen4 and with anti-rabbit Alexa555 second antibody. Images and videos were captured with a Nikon A1 confocal microscope.

**In vivo intracranial xenograft tumor models**. Female SCID mice aged 6–8-weeks were used for intracranial implantation of GBM cells. All animal experiments were carried out under an Institutional Animal Care and Use Committee (IACUC) approved protocol according to NIH guidelines. The mice were housed in groups of five animals per cage and had access to autoclaved water and pelleted feed. The cage environment was enriched with a mouse house. The mice were kept at a standard temperature of 22 ± 2 °C and a relative humidity of 55% (45–70%) in a 12:12-hour light:dark cycle (lights on, 6 a.m. to 6 p.m.). The GBM cells suspension ($2 \times 10^5$ cells in 3 μl of PBS) was injected into the brains of SCID mice as previously described. When neuropathological symptoms developed, mice were sacrificed and perfused with ice-cold PBS and 4% paraformaldehyde (PFA). Mice brains were dissected, fixed in 4% PFA for 24 h and then transferred to 10% formalin.

**In vivo drug treatments**. AR42 (Arno Therapeutics) was dissolved in dimethyl sulfoxide for in vitro treatments or in 0.5% hydroxypropyl methylcellulose (Dow Chemical Company)/0.1% Tween-80 (Sigma) for in vivo administration via oral gavage. On day 10 after injection of GBM cells, mice were administrated with AR42 for 9 days (three times per week, total 3 weeks), and Kaplan–Meier survival curves were generated.

**Lentivirus production and transduction**. Lentiviral vectors expressing shRNA for HDAC1 or CD109 were purchased from Sigma. Plasmid DNA was purified with HiSpeed Plasmid Midi Kit (Qiagen). For lentiviral production, HEK293FT cells were transfected with the vectors (Sigma) and two packaging plasmids (psPAX2 and pMGD2) using the CalPhos Mammalian Transfection Kit (Clontech) according to the manufacturer's protocol. The lentiviral particles were harvested 72 h after transfection and were concentrated 100-fold using a Lenti-X concentrator (Clontech) and stocked −80 °C until infection. One day before infection GBM spheres were dissociated into single cells with accutase and seeded on laminin 6-well plates at $5 \times 10^5$ cells per well. Next day GBM cells were incubated with viral supernatants for 24 h in the presence of 8 µg/ml polybrene (EMD Millipore). At 24 h after infection, medium was renewed and cells were collected at 5 days after infection.

**Co-immunoprecipitation**. Cells were lysed with Lysis Buffer containing 1% protease inhibitor and phosphatase inhibitor. The cell lysis was pre-washed with magnetic beads under 4 °C for 1 h. Then incubate with target antibody at 4 °C overnight. The next day, magnetic beads were added to the antigen sample/antibody mixture and incubated under room temperature for 1 h. One-hundred microliters low-pH elution buffer was used to elute the binding protein. The pH was normalized with 15 µl of neutralization buffer for each 100 µl of elution buffer. IP samples were separated by electrophoresis in 4–12% Bis-Tris protein gel.

**Chromatin immunoprecipitation assay (ChIP)**. The ChIP assay was performed after cross-linking cells using formaldehyde. DNA was sonicated using an Ultrasonic Processor (Bioruptor UCD-200) at 12 cycles of 30 s with 30 s interval between cycles. Sonicated DNA was then centrifuged at $20,000 \times g$ at 4 °C. Supernatant was used for ChIP assay using MAGnify ChIP system (Invitrogen). Two micrograms of mouse IgG, HDAC1 (Abcam-ab15050), H3ac (active motif 39139), C/EBPβ (Genetex GTX100675) was used per ChIP. Immunoprecipitated DNA was analyzed by qPCR, and Ct values were used to calculate the percentage of input enrichment.

**Re-ChIP**. Re-ChIP was performed using a kit from Active Motif (cat.no.53016) according to the manufacturer's protocol.

**CD109 enzyme-linked immunosorbent assay (ELISA)**. The concentration of sCD109 was measured using the CD109 ELISA kit (LSbio) according to the manufacturer's protocol.

**In vivo bioluminescent imaging**. To monitor tumor growth in live animals, GBM spheres were transduced with lentiviral particles (pHAGE PGK-GFP-IRES-LUC-W) for co-expression of GFP and luciferase, and then GFP expressing cells were sorted by FACS. GBM spheres expressing luciferase were intracranially transplanted into immunocompromised SCID mice. To examine the tumor growth, animals were administrated intraperitoneally with 2.5 mg/100 µl solution of XenoLight D-luciferin (PerkinElmer) and anesthetized with isoflurane during the imaging analysis. The tumor luciferase images were captured by using an IVIS100 imaging system (PerkinElmer).

**RNA sequencing**. Briefly, cDNA libraries for paired-end sequencing were prepared using TruSeq Stranded mRNA-Seq Library Preparation Kit (Illumina) according to the manufacturer's protocol. Samples were sequenced with an llumina HiSeq 2500 system (Illumina) and 150 bp paired-end reads were generated. Sequence reads in fastq format were imported into a local instance of galaxy (galaxy.uabgrid.uab.edu). STAR (version 2.5.3a) was used to align the raw RNA-Seq reads to the human reference genome from Gencode (GRCh38 Release 25). Cufflinks was then used on the aligned reads from STAR to assemble transcripts, estimate their abundances and test for differential expression and regulation. Cuffmerge, which is part of Cufflinks, merged the assembled transcripts to a reference annotation and is capable of tracking Cufflinks transcripts across multiple experiments. Finally, Cuffdiff was used for statistical examination of the data.

**Gene expression data analysis**. Heatmap and clustering were performed with Cluster 3.0 software, with results displayed using Java Treeview software. Single-cell RNA sequencing data were downloaded from http://www.gbmseq.org/. Gene Enrichment Analysis was performed using available online software (http://software.broadinstitute.org/gsea/index.jsp). ssGSEA (single-sample GSEA) was applied to determine the IVY regional gene-set enrichment score of paired patient GBM samples. PCA was created with SPSS 22.0.

**LC-MS/MS analysis**. Eight milliliters of cell culture media were concentrated and exchanged in PBS using Amicon Ultra 4 ml 3 kDa NMWL centrifugal unit (Millipore). Approximately 40 µg of protein per sample were then diluted to 35 µl using NuPAGE LDS sample buffer (Invitrogen). Proteins were reduced with DTT and denatured at 70 °C for 10 min prior to loading onto Novex NuPAGE 10% Bis-Tris Protein gels (Invitrogen). The gels were stained overnight with Novex Colloidal

Blue Staining kit (Invitrogen). Following de-staining, each lane was cut into six MW fractions and equilibrated in 100 mM ammonium bicarbonate, each gel plug was then digested overnight with Trypsin Gold, Mass Spectrometry Grade (Promega) following manufacturer's protocol. Peptide extracts were reconstituted in 0.1% Formic Acid/ddH2O at 0.1 µg/µl and used for LC-MS/MS analysis.

**Statistics and reproducibility**. All data are presented as mean ± SD. The number of replicates for each experiment is stated in the figure legend and always refer to independent biological replicates. Statistical differences between two groups were evaluated by two-tailed $t$-test. One-way ANOVA was utilized in comparisons of >2 groups, following Dunnett's Tukey's post-test. The statistical significance of Kaplan–Meier survival plot was determined by log-rank analysis. A statistical correlation was performed to calculate the regression R2 value and Pearson's correlation coefficient. Statistical analysis was performed by Prism 6 (Graphpad Software), unless mentioned otherwise in figure legend. $p < 0.05$ was considered statistically significant. Exact $p$-values are provided in Supplementary Data 6.

**Reporting summary**. Further information on research design is available in the Nature Research Reporting Summary linked to this article.

## Data availability
RNAseq data of 12 GBM sphere lines and 9 tissue samples have been deposited at the Gene Expression Omnibus database under the accession code GSE153746. Gene expression data from Ivy Glioblastoma Atlas Project [https://www.ncbi.nlm.nih.gov/geo/query/acc.cgi?acc=GSE107560] and GBMseq [https://www.ncbi.nlm.nih.gov/geo/query/acc.cgi?acc=GSE84465] were used in this study, as well as expression/survival data from the Repository for Molecular Brain Neoplasia [https://www.ncbi.nlm.nih.gov/geo/query/acc.cgi?acc=GSE108476]. All the other data supporting the findings of this study are available within the article and its supplementary information files.

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

## Acknowledgements

We would like to thank members of Dr. Nakano's lab at the University of Alabama (UAB) at Birmingham for input on manuscript writing and editing. We thank Dr. Harley I. Kornblum (M.D., Ph.D.) at the University of California, Los Angeles for sharing GBM spheres (157, 336, 339, 374, and 408). We would like to thank the UAB High Resolution Imaging facility, Comprehensive Cancer Center bio-imaging core and proteomics core service teams, funded by National Institute of Health (NIH); National Cancer Institute (NCI), project # P30CA013148, and also the UAB Institutional Core Funding Mechanism. MSP was supported by the Russian Foundation for Basic Research grant 20-34-70147 and Russian Science Foundation grant 19-44-02027 (LC-MS/MS analysis and RNA sequencing). I.N. was supported by NIH grants: R01NS083767, R01NS087913, R01CA183991 and R01CA201402. A special thanks to Joseph R. Gould (Department of Microbiology, UAB) for his diligent editing and proofreading of this manuscript.

## Author contributions

I.N. designed the overall experiments and analyzed data. I.N., M.S.P., and S.B. wrote the manuscript. S.B., M.S.P., Z.Z., M.M., S.B., D.Y., S.G., H.S., S.Y., S.K., N.K., E.Y., J.M., K.S., K.B., K.S.Y., and Q.C. performed cell culture and/or animal experiments. D.K.C., D.Y., S.Y., Y.L., and H.C. performed bioinformatics analysis of published expression datasets. E.Y., K.B., J.M., L.B.N., J.L., I.N., and D.H.N. provided intellectual input. All authors provided scientific input, edited, and approved the final manuscript.

## Competing interests

The authors declare no competing interests.
