## [Peer Review File · Nature Communications]

Reviewers' comments:

Reviewer #2 (Remarks to the Author); expert in glioblastoma, mouse models, therapy:

Yu et al. propose a model of intratumoral, intercellular communication in glioblastoma (GBM) between tumor cells located in the core of the tumor and tumor cells located at the invasive tumor edge.

Major points:

(1.) The vast majority of data is derived from in-vitro co-culture and in-vivo co-injection of distinct GBM cell populations. It is not clear what fraction of human GBMs, and perhaps which of the three main molecular GBM subgroups, harbor these two types of glioma stem-like cells. This cannot be addressed through bulk RNAseq data from publically available datasets. Instead, the coexistence of these distinct cell populations and their (presumed) cell type specific signaling molecules needs to be documented through in-situ staining of a larger panel of human GBMs.

(2.) The authors previously reported that the invading/infiltrative edge of human GBMs are comprised of different glioma stem-like cells (CD109+) than the tumor core (CD133+), resembling the mesenchymal and proneural GBM subgroups, respectively (Minata et al., Cell Rep 2019). The current study extends these findings by suggesting that CD109 expression is transcriptionally regulated by HDAC1 and potentially further refines our understanding of the contribution of class I HDACs in a subset of brain-tumor initiating cells (PMID: 27449082). If the authors' model is correct, inhibition of HDAC1 should selectively inhibit the growth of GBM xenografts of the mesenchymal subgroups (presumably derived from CD133+ GSCs) compared to GBM xenografts of the proneural GBM subgroup (presumably derived from CD109+ GSCs). This should be shown in a larger panel of models, using MRI-based in-vivo tumor volume measurements.

(3.) The cell lines used in this study should be described in greater detail (authentication, genetics, etc), as they appear to be different from the authors' prior publication focusing on the same phenotypic classification of "tumor core"-like versus "invading edge"-like glioma like stem cells.

Reviewer #3 (Remarks to the Author); expert in HDAC inhibition:

This manuscript presents a wealth of data which characterizes the role of the GBM core and edge cells. It begins with a description of the transcriptional signature for GBM cells at the tumor core vs tumor edge. CD109 in its soluble form is identified as signal from the core promotes growth and resistance to radiation of edge cells. HDAC1 is identified via small molecule, inhibitor, and shRNA studies as the initiator of signaling from core to edge cells. To identify the transcriptional pathway involved, interaction between C/EBP β HDAC1 is confirmed by ChIP but the involvement of STAT3 is mentioned but not shown here. As the manuscript describes the validation of the in vitro findings developed with patient derived samples with animal studies and then correlates these results with patient data and outcomes, the authors validate their data on many relevant levels. Certainly the multiple approaches used to support the conclusions of the paper is a key strength and the paper appears to have no key issues with validity. However, there are flaws in presentation and the lack of context which detracts from the impact of the results and limits the audience for this work.

Most importantly, the presentation of the data is extremely lacking in clarity and this detracts from its impact. Most data presented is repeated in multiple, repetitive panels. The authors should take the time to consolidate the data from the multiple panels into one – or provide a rationale to include multiple panels beyond the often repeated "Fig Xa, b and Supplemental Fig. Xa, b". It appears that the figures were developed by multiple laboratories and simply collated. Without any

legends to the supplemental figures and less than satisfactory legends in the figures in the body of the manuscript, it is unclear how and why each point is shown multiple ways. If this repetition is necessary, the need for it needs to be explained and supplemental figures need to be related to those included in the manuscript. It's clearly the responsibility of the authors to more completely interpret and present their data in a more cohesive fashion.

The three brief paragraphs entitled "Ethics", "Statistical Analysis", and "Data Availability" under the methods section do not allow evaluation of the methods used and do not allow for reproduction of results. Thus the methods section is woefully inadequate. When the authors take the time to consolidate figures, it will be easier to evaluate the statistical analysis for each figure. However, there are panels where ANOVA is the appropriate test and this analysis is not mentioned in the methods section.

In addition to the more coherent presentation of the data, points to address in any proposed revision to this or any other journal:

1. The manuscript would benefit from an extensive reorganization. Figure 1 is first used to describe how the core and edge cells are obtained – something which appears to have been published previously by some of the authors (indeed CD133 has been previously described as an edge cell marker even though the full transcriptional profiling may be new). The derivation is then addressed again at the end of the manuscript. This repetition is not necessary and results in confusion.
2. The authors state that the harsh environment in the core produces is expected to produce the cells invade surrounding tissues and which lead to recurrence (568-571). This has never been my understanding from the moment I read my first cancer biology textbook.
3. Proteinase K experiments do not establish that the CD109 isn't bound as a GPI anchored protein on vesicles in conditioned media. However,
4. There is nothing in the introduction to contextualize CD109 and its role in cancer biology. Likewise with CD133 although this is not as essential. I also suggest inclusion in the discussion how sCD109 relates to previous ECV results.
5. I would like to see some indication of the specificity of the HDAC inhibitors used in this study as inhibitors targeting HDACs have been shown to affect the expression of a large percentage of the genome.

In conclusion, the manuscript presents a plethora of data and the different levels (e.g. in vitro, mouse model, and human patient) of data which are used to weave a coherent story is a major strength of the manuscript. However, but suffers from major flaws which detract – even destroy – the impact of the data presented. The authors need to collaborate to assemble a impactful and coherent presentation of the data or separate the data presented into multiple smaller manuscripts. Please keep in mind that my expertise is in vitro work and, although we do work with human tissues and primary cultures, my laboratory does not work with mouse models.

Reviewer #4 (Remarks to the Author); expert in tumour microenvironment:

This is a wide ranging and ambitious study on the highly relevant topic of infiltrative growth of GBM.

However, in its present form the study is unfortunately flawed by some conceptual vagueness and a design including many different experiments that are not well integrated (Figs. 2-6 vs Fig. 1,7-8).

The reliance on analyses of patient-derived cell models are a merit of the study. However, clinical relevance of model studies is not, in this version, well supported by detailed correlative studies of human tissue.

Findings of Fig. 7 and 8 appear most promising, original and relevant.

Authors are encouraged to further develop these findings.

Main points:

1. The study should better describe if the aim is to understand properties of the infiltrative GBM cells that are left in setting of complete resection, or if it aims to understand "intra-tumoral" biology including communications between central and peripheral regions of contrast-enhancing tumor tissue. The former appear as an original and important study topic. The significance and relevance of the latter is less obvious.

As the study presently lacks this distinction it remains difficult to evaluate relevance of model studies and to suggest the proper correlative analyses.

2. Studies of Fig. 2-6, concluding with a model of core-edge signaling including HDAC1-regulated expression of CD109 in core cells, rely exclusively of analyses using "core-like" and "edge-like" spheres. These are defined by expression of CD133 and CD109, respectively, without considering their original localization. As of now authors does not convincingly show relevance of these models for the spatially distinct GBM stem cells they aim to describe.

3. Studies of Fig. 7 and 8 provides some preliminary and interesting findings. The demonstration of differences in core- and edge-derived cultures regarding gene expression profiles (7a, b) and phenotypic properties (migration,, invasion, in vivo growth (7c-g)) appear very promising. Furthermore, some functional complementarity is demonstrated in cell- and mouse-studies (Fig. through co-injection studies (Fig. 8 a, b). However, with the exception of CHIP-based demonstration of HDAC1 binding to CD109 promotor in core cells (Fig. 8h), these studies lack molecular analyses. Notably, this particular experiment is lacking relevant and expected controls which seriously limits the significance.

Point by point reply to the reviewers' comments

We have added significant amount of new data as detailed below. Briefly, the major changes that we have made are:

1. We have performed IHC staining of 61 clinical GBM samples with Olig2 and CD109.
2. We have compared the sensitivity of edge-like and core-like GBM spheres to HDAC inhibitor (AR42) *in vivo*.
3. We have added a supplementary table for description of details of sphere lines used in this study.
4. We have tested the concentration of CD109 in conditioned media with/without extracellular vesicles.
5. We have performed additional molecular analysis for paired core and edge GBM spheres, including ones with knockdown and overexpression of HDAC1 and also performed ChIP of HDAC1 in edge and core spheres.
6. We have carefully modified the main text, to make it easier for understanding.

Reviewer #2 (Remarks to the Author); expert in glioblastoma, mouse models, therapy:

Yu et al. propose a model of intratumoral, intercellular communication in glioblastoma (GBM) between tumor cells located in the core of the tumor and tumor cells located at the invasive tumor edge.

Major points:

(1.) The vast majority of data is derived from in-vitro co-culture and in-vivo co-injection of distinct GBM cell populations. It is not clear what fraction of human GBMs, and perhaps which of the three main molecular GBM subgroups, harbor these two types of glioma stem-like cells. This cannot be addressed through bulk RNAseq data from publically available datasets. Instead, the coexistence of these distinct cell populations and their (presumed) cell type specific signaling molecules needs to be documented through in-situ staining of a larger panel of human GBMs.

Reply: We thank reviewer for this comment. We have performed IHC staining of 61 GBM samples (PN, CL and MES subtypes) with Olig2 (presumed edge specific) and CD109 (presumed core specific) using our tissue microarray (TMA). We found that more than half of the GBM samples demonstrated coexistence of these markers (38 out of 61). The result showed that Olig2⁺ and CD109⁺ tumor cells co-existed in all three subtypes at variable ratio (Supplementary Fig. 1e). Indicating that edge and core signature may be relevant to all subtypes of GBM. In the PN subtype GBM, Olig2 showed higher staining intensity and CD109 demonstrated relative lower and variable intensity. In the MES subtype, CD109 showed higher intensity and Olig2 demonstrated relative lower and variable intensity. (Supplementary Fig. 1f).

Supplementary Fig. 1f. Heatmap showing the staining intensity of Olig2 and CD109 in all three subtypes of GBM.

Supplementary Fig. 1e. Representative IHC staining of PN and MES GBM tumor tissues for Olig2 and CD109.

(2.) The authors previously reported that the invading/infiltrative edge of human GBMs are comprised of different glioma stem-like cells (CD109+) than the tumor core (CD133+), resembling the mesenchymal and proneural GBM subgroups, respectively (Minata et al., Cell Rep 2019). The current study extends these findings by suggesting that CD109 expression is transcriptionally regulated by HDAC1 and potentially further refines our understanding of the contribution of class I HDACs in a subset of brain-tumor initiating cells (PMID: 27449082). If the authors' model is correct, inhibition of HDAC1 inhibition should selectively inhibit the growth of GBM xenografts of the mesenchymal subgroups (presumably derived from CD133+ GSCs) compared to GBM xenografts of the proneural GBM subgroup (presumably derived from CD109+ GSCs). This should be shown in a larger panel of models, using MRI-based *in-vivo* tumor volume measurements.

Reply: We have injected the mice with 1051 (edge-like, CD133⁺) and 267 (core-like, CD109⁺) sphere lines separately and treated them with vehicle or AR42 (HDACs inhibitor). Unfortunately, we don't have the facility for MRI-based *in vivo* tumor volume measurements. Therefore, the *in vivo* response of tumor cells to the inhibitor was monitored with luciferase-based bioluminescence imaging. The result show that the core-like GBM cells were more sensitive to HDAC inhibitors, compared with edge-like counterpart (Supplementary Fig. 5a).

Supplementary Fig. 5a. Representative bioluminescence imaging of mice injected with edge-like 1051/core-like 267 and treated with vehicle or AR42.

Figure 5d. Quantification of BLI signal. ***P<0.01 using Student's t-test. Data are mean \pm s.d.

(3.) The cell lines used in this study should be described in greater detail (authentication, genetics, etc), as they appear to be different from the authors' prior publication focusing on the same phenotypic classification of "tumor core"-like versus "invading edge"-like glioma like stem cells.

Reply: We thank for the reviewer for this comment. We have added a table (Supplementary table 1) with information regarding cell lines that were used in this study. It includes patient details, STR analysis and marker expression.

Reviewer #3 (Remarks to the Author); expert in HDAC inhibition:

This manuscript presents a wealth of data which characterizes the role of the GBM core and edge cells. It begins with a description of the transcriptional signature for GBM cells at the tumor core vs tumor edge. CD109 in its soluble form is identified as signal from the from the core promotes growth and resistance to radiation of edge cells. HDAC1 is identified via small molecule, inhibitor, and shRNA studies as the initiator of signaling from core to edge cells. To identify the transcriptional pathway involved, interaction between C/EBP β HDAC1 is confirmed by ChIP but the involvement of STAT3 is mentioned but not shown here. As the manuscript describes the validation of the in vitro findings developed with patient derived samples with animal studies and then correlates these results with patient data and outcomes, the authors validate their data on many relevant levels. Certainly the multiple approaches used to support the conclusions of the paper is a key strength and the paper appears to have no key issues with validity. However, there flaws in presentation and the lack of context which detracts from the impact of the results and limits the audience for this work. Most importantly, the presentation of the data is extremely lacking in clarity and this detracts from it's impact. Most data presented is repeated in multiple, repetitive panels. The authors should take the time to consolidate the data from the multiple panels into one – or provide a rationale to include multiple panels beyond the often repeated "Fig Xa, b and Supplemental Fig. Xa, b". It appears that the figures were developed by multiple laboratories and simply collated. Without any legends to the supplemental figures and less

than satisfactory legends in the figures in the body of the manuscript, it is unclear how and why each point is shown multiple ways. If this repetition is necessary, the need for it needs to be explained and supplemental figures need to be related to those included in the manuscript. It's clearly the responsibility of the authors to more completely interpret and present their data in a more cohesive fashion.

Reply: We thank reviewer for this comment and apologize for the poor presentation of the data in the original version of the manuscript. To make a more coherent presentation of our results, we have carefully rewritten the manuscript to make sure all context information were added and the description of result was clear enough. We have improved the legends and confirmed that they were placed with corresponding figures. Finally, we added a scheme demonstrating the hypothesis proposed in this study (Fig. 8f).

Fig. 8f. Proposed molecular mechanism of intercellular crosstalk between core and edge GBM cells via soluble CD109 protein. This signaling induces upregulation of CD44 and cMyc in edge cells and ultimately leads to increased

The three brief paragraphs entitled “Ethics”, “Statistical Analysis”, and “Data Availability” under the methods section do not allow evaluation of the methods used and do not allow for reproduction of results. Thus, the methods section is woefully inadequate. When the authors take the time to consolidate figures, it will be easier to evaluate the statistical analysis for each figure. However, there are panels where ANOVA is the appropriate test and this analysis is not mentioned in the methods section.

Reply: The detailed description of all used methods was included in the main text of the manuscript. We have double checked the statistical analysis for each figure. We have added the description for ANOVA analysis and stated where it was applied.

In addition to the more coherent presentation of the data, points to address in any proposed revision to this or any other journal:

1. The manuscript would benefit from an extensive reorganization. Figure 1 is first used to describe how the core and edge cells are obtained – something which appears to have been published previously

by some of the authors (indeed CD133 has been previously described as an edge cell marker even though the full transcriptional profiling may be new). The derivation is then addressed again at the end of the manuscript. This repetition is not necessary and results in confusion.

Reply: We thank reviewer for this helpful suggestion. We have reorganized the manuscript using the following story flow: Fig.1 – Description of edge and core GBM tissues and introduction the novel concept of edge-core heterogeneity in GBM; Fig.2 – Description of core and edge derived GBM sphere lines that recapitulate edge/core phenotypes *in vitro* and *in vivo*; Fig.3 – Investigation of edge/core intercellular crosstalk using edge-like and core-like spheres; Fig.4. - Confirmation of data in Fig.3 using our original edge and core sphere models; Fig.5. – Describing HDAC1 as a molecule that is partly responsible for core phenotype; Fig.6. – HDAC1 plays a role in intercellular communication between core and edge GBM cells; Fig.7. – Soluble CD109 is a mediator of HDAC1-derived intercellular signals from core to edge GBM cells; Fig.8. – Expression of CD109 is directly regulated by HDAC1*CEBPb complex.

2. The authors state that the harsh environment in the core produces is expected to produce the cells invade surrounding tissues and which lead to recurrence (568-571). This has never been my understanding from the moment I read my first cancer biology textbook.

Reply: Indeed, during initial tumor grow tumor expands and later tumor core with necrosis is formed due to the lack of blood supply. However, in this paper, we studied the process of tumor recurrence after surgery, when most (but not all) of the core was removed and remaining relatively small population of core cells affect relatively intact population of edge cells. We have clarified it in the discussion section: “Clinically, it is obvious that some, if not all, edge-located tumor cells give rise to new core lesions at recurrence. This edge-core transition was presumably partially accompanied by the post-radiation PN-MES transition in tumor-initiating cells, which we previously identified in GBM 13. However, these concepts provoke an unresolved question. Owing to the lack of a reliable experimental recurrence model elucidating the evolutionary processes undertaken throughout GBM microtumor development and core re-formation, we cannot fully capture the array of pro-tumorigenic signaling events for tumor re-establishment throughout recurrence development. We are underway to address this question.”

3. Proteinase K experiments do not establish that the CD109 isn't bound as a GPI anchored protein to vesicles in conditioned media.

Reply: To address this issue used filtration to remove vesicles and subsequently measured the concentration of CD109 in media. The result showed that filtration didn't significantly altered the concentration of CD109 in conditioned media (Supplementary Fig. 7d).

Supplementary Fig. S7d. Enzyme-linked immunosorbent assay (ELISA) of secreted CD109 in CM from core-like 267 GBM spheres with/without filtration. ns, not significant.

4. There is nothing in the introduction to contextualize CD109 and its role in cancer biology. Likewise, with CD133 although this is not as essential. I also suggest inclusion in the discussion how sCD109 relates to previous ECV results.

Reply: We apologize for the lacking of information. We have added the information about CD133 and CD109 to the introduction section: “One of the few such markers include CD133 and CD109. CD133 (Prominin-1), a cell surface glycoprotein expressed on neural precursor cells, is a well-known marker for identification of glioma-initiating cell subpopulation among others (Clin Transl Med. 2018 Jul 9;7(1):18). Cell surface expression of CD133 appears to be correlated with tumor initiating property following the current first-line post-surgical therapies (Radiol Oncol. 2011 Jun;45(2):102-15). Of note, CD133 positive cells were shown to be enriched in the infiltrating edge of GBM tumors (Oncol Lett. 2019 Feb;17(2):2123-2130) On the other hand, CD109, a glycosylphosphatidylinositol-anchored glycoprotein, is highly expressed in multiple tumors and associated indecently with worse outcome of patients. In glioma, CD109 was found to be closely linked with tumor-initiating cells. Most recently, our study further demonstrated an association of CD109 with the signature of GBM cells located at the core of the tumor (Cell Rep. 2019 Feb 12;26(7):1893-1905). However, the applicability of these two markers for spatial identity has not been systematically validated.”

5. I would like to see some indication of the specificity of the HDAC inhibitors used in this study as inhibitors targeting HDACs have been shown to affect the expression of a large percentage of the genome.

Reply: For the HDACs inhibitors used in the screening, we have indicated their specificity in the Figure 5A. Besides, we have discussed the specificity of AR42 (HDAC inhibitor) in the discussion section. “Clinically, our results indicated that specific inhibition of HDAC1 is a potential strategy for future combination treatment of GBM after surgical resection. However, the current HDACs inhibitors in clinical trials are targeting multiple different HDACs, like vorinostat, trichostatin A or panobinostat, which target class I, II and IV HDACs. To confirm the pharmacological inhibition effect of HDAC1, we used AR42, a class I and class II HDACs inhibitor, in our study. Thus, developing and investigation of HDAC1 specific inhibitor will help with validation of our study and potentially contribute to the future clinical treatment.”

In conclusion, the manuscript presents a plethora of data and the different levels (e.g. in vitro, mouse

model, and human patient) of data which are used to weave a coherent story is a major strength of the manuscript. However, but suffers from major flaws which detract – even destroy – the impact of the data presented. The authors need to collaborate to assemble a impactful and coherent presentation of the data or separate the data presented into multiple smaller manuscripts. Please keep in mind that my expertise is in vitro work and, although we do work with human tissues and primary cultures, my laboratory does not work with mouse models.

We thank again for the positive and helpful comments raised by reviewer 3.

Reviewer #4 (Remarks to the Author); expert in tumour microenvironment:

This is a wide ranging and ambitious study on the highly relevant topic of infiltrative growth of GBM. However, in its present form the study is unfortunately flawed by some conceptual vagueness and a design including many different experiments that are not well integrated (Figs. 2-6 vs Fig. 1,7-8).

The reliance on analyses of patient-derived cell models are a merit of the study. However, clinical relevance of model studies is not, in this version, well supported by detailed correlative studies of human tissue.

Findings of Fig. 7 and 8 appear most promising, original and relevant.

Authors are encouraged to further develop these findings.

Main points:

1. The study should better describe if the aim is to understand properties of the infiltrative GBM cells that are left in setting of complete resection, or if it aims to understand “intra-tumoral” biology including communications between central and peripheral regions of contrast-enhancing tumor tissue. The former appear as an original and important study topic. The significance and relevance of the latter is less obvious. As the study presently lacks this distinction it remains difficult to evaluate relevance of model studies and to suggest the proper correlative analyses.

Reply: We thank reviewer for this important comment. Our aim is to investigate the intercellular crosstalk from the leftover contrast-enhancing tumor tissue (core) to the infiltrating tumor cells (edge, T2 FLAIR). We have included a clear description for our aim in the introduction section “In this study, we aimed at bringing out the concept of recurrence initiating cells (RICs) and investigation of the intercellular crosstalk between surgical leftover core tumor cells and RICs.”.

We agree that edge cells are of great importance for the post-surgery treatment and tumor recurrence. However, we also want to claim that the leftover core cells due to incomplete resection are playing a key role in the process. First, the incomplete resection rate is extremely high in current clinical scenario according to our analysis of recent 17 studies within 8 years (Figure 3A). Second, our and other studies have showed that cell cross-talk contributed a lot to treatment resistance and tumor recurrence (Cancer research 78, 3002-3013; Cancer cell 34, 119-135). Third, recent study shows that distinct tumor cell populations contribute to the heterogeneity of GBM (Science, 344, 1396-1401). According to Patel’s single cell study, the more heterogeneous tumor, the worse the patient’s outcome. Study of Anne also revealed that more plastic states of GBM accelerated tumor growth of GBM (Nat Commun. 2019 Apr 16;10(1):1787). We believe that co-existence of edge and core make the outcome of patients much worse.

2. Studies of Fig. 2-6, concluding with a model of core-edge signaling including HDAC1-regulated expression of CD109 in core cells, rely exclusively of analyses using “core-like” and “edge-like” spheres. These are defined by expression of CD133 and CD109, respectively, without considering their

original localization. As of now authors does not convincingly show relevance of these models for the spatially distinct GBM stem cells they aim to describe.

Reply: We included data to demonstrate that the core-like and edge-like sphere model closely resembles regionally-defined core and edge GBM cells. In addition, we repeated several HDAC related experiments using original core and edge GBM cells (Fig. 7g and Fig. 8c).

A table on Supplementary Fig. 3f showing the characteristic of both edge/edge-like and core/core-like GBM spheres and the original tissues.

Characteristic of tissues and glioma spheres						
	Transcriptional signature			Malignant biological behavior		
		markers	pathways	Infiltration	location preference	IR-resistance
Tissue	edge	Olig2	KRAS	-	-	-
	core	CD44/CD109	c-MYC/G2M check point	-	-	-
Spheres	edge-like	CD133	KRAS	high	edge	low
	edge	CD133	KRAS	high	edge	low
	core-like	CD109	c-MYC/G2M check point	low	core	high
	core	CD109	c-MYC/G2M check point	low	core	high

On the transcriptional profile level, we showed that edge-like and edge, core-like and core GBM cells possessed similar gene signature (Fig. 2f) and pathway alternation (Fig. 2g) to the samples that were obtained from clinic. On the *in vivo* behavior aspect, we showed that edge-like and edge GBM cells were much more infiltrative and preferentially formed edge part in co-injection xenograft (Fig. 2a, 2e and S3e). Furthermore, we found that the core-like and core GBM cells were much radioresistant (Fig. 4c and S4g) and promoted the cell growth (Fig. 4a, 4b, 3c and 3d) and radioresistance (Fig. 4d, 4g, and 3e) of edge-like and edge counterparts. These results indicated that though the edge-like and core-like model are of limitation, they demonstrated lots of similarity with the matched edge and core spheres and GBM samples. We agree that utilizing membrane marker CD133 and CD109 solely is of limitation for defining of edge and core tumor cells in GBM. To further avoid confusion and make it clear for the future audience, we added a clearer description before applying this sphere model, “First, we tested the expression levels of previously identified markers of edge (CD133 and Olig2) and core (CD109) (Oncol Lett. 2019 Feb;17(2):2123-2130; Cell Rep. 2019 Feb 12;26(7):1893-1905) in GBM spheres derived from 12 different patients (Supplementary Fig. 3a and 3b). Results of this experiment allowed us to subdivide sphere lines on “core-like” (20, 1005, 1020, 267) and “edge-like” (157, 711, 1027, 1051, 1079). Consistent with these data, the regionally derived edge spheres demonstrated higher CD133 expression and lower expression of CD109 and CD44, compared with the core counterparts (Supplementary Fig. 3c and 3d). Next, we performed RNA-seq of edge/core and edge-like/ core-like spheres. PCA analysis demonstrated significant similarity in gene expression of edge and core spheres with the previously-established edge-like and core-like spheres, respectively (Fig. 2f and Supplementary table 1). Of note, GSEA exhibited that c-Myc and G2M checkpoint were among the top differentially-upregulated pathways in the core or core-like lines, while KRAS was identified in the edge or edge-like lines (Fig. 2g).”

3. Studies of Fig. 7 and 8 provides some preliminary and interesting findings. The demonstration of differences in core- and edge-derived cultures regarding gene expression profiles (7a, b) and phenotypic properties (migration, invasion, *in vivo* growth (7c-g)) appear very promising. Furthermore, some functional complementarity is demonstrated in cell- and mouse-studies (Fig.

through co-injection studies (Fig. 8 a, b). However, with the exception of CHIP-based demonstration of HDAC1 binding to CD109 promotor in core cells (Fig. 8h), these studies lack molecular analyses. Notably, this particular experiment is lacking relevant and expected controls which seriously limits the significance.

Reply: We have performed additional experiments to confirm our hypothesis, including silencing HDAC1 in core sphere cells, overexpressing HDAC1 in edge sphere cells (Fig. 7g) and ChIP of HDAC1 in both core and edge sphere cells (Fig. 8c). Results of these experiments have demonstrated that the silencing of HDAC1 in 1051 core GBM cells significantly reduced the expression of CD109 while the overexpression of HDAC1 in 1051 edge GBM cells upregulated CD109 expression. In agreement with these data, we demonstrated much higher occupancy of HDAC1 on CD109 promoter in core cells as opposed to edge cells. All together, these results further confirm the regulation of CD109 expression by HDAC1.

Figure 7g. WB for HDAC1 and CD109 in 1051 core GBM cells infected with shNT or shHDAC1 viruses and in 1051 edge GBM cells infected with GFP or HDAC1 encoding viruses.

Figure 8c. ChIP analysis showing binding of HDAC1 to CD109 promoter region in 1051 core and edge GBM spheres. *** $P < 0.001$ using Student's t-test.

Reviewers' comments:

Reviewer #2 (Remarks to the Author):

The authors have substantially revised the manuscript but concerns remain regarding the robustness of key conclusions.

For example, the authors examined staining intensities for Olig2 and CD109 using an arbitrary 0/1/2/3 scale. It would seem more relevant to report the percentage of tumor cells staining for each of these markers and also the percentage of tumor cells co-labeling with these markers. Given the central hypothesis (i.e., intratumoral signaling between core-like cells and edge-like cells), it also seems that a more extensive spatial examination of human GBMs (e.g., multiple FFPE blocks, sampled with detailed regional annotation) would be far more informative than the use of tumor microarrays which typically only represent a tiny fraction of each tumor.

Similarly, the conclusion of differential tumor sensitivity to AR42 is based on only one experimental model per subgroup (1051 cell line for "edge-like" GBM cells and 267 cell line for "core-like" GBM cells) and there is considerable variability even within the group of vehicle-treated mice.

Reviewer #3 (Remarks to the Author):

The addition of the data stemming from the addition of supplementary Fig. 5a in response to Major point 2 by Reviewer #2 improves the manuscript. However, there are several more points which should be addressed:

The new data in supplementary data 1f should be quantitated and statistically evaluated. I suspect that with the sample size ($n=61$ with 38 positive for both oligo and CD109 expression) divided into four groups (PN, CL, UK, and MES), will not be under-powered and thus the data presented will not be significant. However, this needs to be stated so that these data can be put into appropriate context for the reader.

The term "HDAC1 signaling" is a bit of a misnomer as it is mainly a transducer through interactions with multiple other proteins and lysine acetylation of many non-histone proteins to impact a myriad of pathways. As most HDAC inhibitors affect transcription of approximately one third of the genome (with gene expression levels increased and decreased at almost equal frequencies), it is not unexpected that 1) CD109 expression is increased downstream of HDAC1 nor 2) CD109 promoter occupancy by HDAC1 and C/EBP β is low. However, the fact that it is not as novel as the authors' posit, does not detract from their findings.

The authors' presentation of other HDAC inhibitors in clinical trials is a little misleading as well. There are a total of 10 HDACs in classes 1 and 2 and a single HDAC in class IV. Thus, it is difficult to claim AR-42 is significantly more specific than other HDAC inhibitors listed – particularly with the lack of established IC50s for AR-42 (lines 512-514).

Reviewer #4 (Remarks to the Author):

The overall concept promoted by the study, now better defined in the "graphical abstract" (Fig. 8F), is the existence of a distinct edge-localized GBM cell population which acts as recurrence-initiating cells (RICs) based on a set of properties of which some are cell-autonomous whereas others rely on paracrine signaling from post-surgery-remaining GBM core cells.

Also in the revised improved version the most interesting and translationally relevant findings are those that rely on the regionally defined GBM sub-populations. The summary below, and remaining major issues, are therefore mostly focusing on the experiments using those cell models and on the tissue analyses.

The edge RICs display certain gene-expression and GSEA profiles (Fig. 1 A, B, C, E) consistent with earlier published studies (Fig. 1 D). Furthermore sphere cultures of edge and core cells display, after brain injection, some properties consistent with their original counterparts including gene expression and pathway activation (Fig. 2 B, F,G). Edge cells show larger invasive capacity in vivo and in vitro (Fig. 2A, C). Furthermore, core cells are claimed to enhance some edge cell phenotypes, including invasion in vivo and in vitro (Fig. 2D,E), cell and tumor growth, IR resistance and CD44 expression (Fig. 4). The paracrine capacity of edge cells are suggested to depend on HDAC1 based on shRNA and inhibitor experiments in studies that uses different endpoints including in vitro and in vivo cell growth (Fig. 6A-D) and molecular properties such as overall gene expression, c-myc expression, G2/M checkpoint (Fig. 6 E-G). Finally, a set of studies are shown to make the claim that C/EBPbeta-dependent production of CD109 by core cells are the HDAC1-dependent molecular mediators of the edge cell-derived paracrine capacities (Figs. 7 G, I, J; 8 C).

Major points

1. Fig 2A, B, C, F and G suggest stable cell autonomous differences between core and edge cells in analyses with endpoints including in vivo and in vitro invasion (2A, C) and c-myc, K-RAS, cell cycle status (2B, G). In contrast, experiments with the same models in Figs. 2D/E, 4, 6, 7 and 8 emphasize the strong dependence of edge cell phenotypes (cell growth, IR resistance and CD44 expression) on core cell-derived paracrine signaling. Notably, the cell autonomous c-myc and G2/M checkpoint phenotype of edge cells (Fig 2B, G) are in Fig. 6 E, F shown to be controlled by HDAC1-dependent paracrine signaling (Fig. 6E, F). This inconsistency should be resolved.
2. Authors are encouraged to more clearly highlight the fact that some edge cell phenotypes appear to be cell autonomous whereas other rely on edge cell-derived paracrine signaling. This should be clear in abstract and possibly also reflected in the title of the study.
3. In this perspective it is notable that the infiltrative growth capacity (Fig. 2A, C) appear to be a cell autonomous property of edge cells. This raises some concerns regarding the therapeutic option, implied by the study, to block recurrences by interfering with paracrine edge/core cell signaling. This concern should be highlighted in the Discussion.
4. Regarding the HDAC-C/EBPbeta-CD109 pathway, analyses still are missing for the regionally derived models regarding critical endpoints such as cell growth and IR-resistance.
5. The study is still largely lacking correlative data from human tissue analyses to validate the HDAC-C/EBPbeta-CD109 pathway. Quality of study would be significantly improved if this pathway could be mapped in tissue sections using multiplex-antibody-profiling including pathway components and edge- and core-cell markers. Since authors imply the existence of edge cells in surgically removed tissue (Fig. 3) these studies could be possible to do in regular GBM specimens.

Minor points

1. Findings of Fig. 3A are interesting and provides an important rationale for the study. Possibly it can therefore be introduced as Fig. 1 A.
2. Yellow star is missing in right panel of Fig. 2A.
3. The text below graphs of Fig. 2 G are not clear. They do not appear to indicate the groups that are compared in the GSEA analyses. If not; what do they indicate?
4. In Fig. 3D mouse pictures and graphs do not match; middle picture refers to top graph and top picture refers to middle graph
5. Text should be checked; e.g. "respectable tumor cell" (line 522) should probably be "resectable tumor cells".

Point by point reply to the reviewers' comments

Reviewer #2 (Remarks to the Author):

The authors have substantially revised the manuscript but concerns remain regarding the robustness of key conclusions.

1) For example, the authors examined staining intensities for Olig2 and CD109 using an arbitrary 0/1/2/3 scale. It would seem more relevant to report the percentage of tumor cells staining for each of these markers and also the percentage of tumor cells co-labeling with these markers.

Reply: We thank reviewer for this comment. We used a standard arbitrary scale to facilitate subsequent data quantification. Score “0” indicate less than 1% of stained cells in the sample, “1” – 1-30%, “2” – 30-70% and “3” more than 70% of stained cells. We included examples of staining for both Olig2 and CD109 to illustrate how the corresponding scores were assigned. Less than 5% of cells were stained for both Olig2 and CD109 (pleases see reply to the next question).

Fig.S1 e Representative IHC staining of human GBM tissues for Olig2 (left) and CD109 (right) illustrating different staining intensities. Scale bar 200 μ m.

2) Given the central hypothesis (i.e., intratumoral signaling between core-like cells and edge-like cells), it also seems that a more extensive spatial examination of human GBMs (e.g., multiple FFPE blocks, sampled with detailed regional annotation) would be far more informative than the use of tumor microarrays which typically only represent a tiny fraction of each tumor.

Reply: We agree with this important comment. To address this issue we used immunofluorescent staining and subsequent confocal microscopy to investigate distribution of Olig2⁺ cells and CD109⁺ cells within the tumor. As expected, we demonstrated that the tumor core is enriched for CD109⁺ cells, while tumor edge from the same patient contains more Olig2⁺ cells. Importantly, only a few cells showed simultaneous staining for both markers and it is possible that these cells

appear due to the unspecific staining with both secondary antibodies rather than due to the simultaneous expression of both markers.

Fig.1 f Representative immunofluorescent staining of edge and core human GBM tissues for Olig2 (green), CD109 (red) and DNA (blue). Scale bar 50 μ m.

3) Similarly, the conclusion of differential tumor sensitivity to AR42 is based on only one experimental model per subgroup (1051 cell line for "edge-like" GBM cells and 267 cell line for "core-like" GBM cells) and there is considerable variability even within the group of vehicle-treated mice.

Reply: To address this issue we tested the effect of AR42 on paired edge/core-derived GBM (neuro)spheres obtained from two different patients. According to our data, in both cases, core cells were more sensitive to AR42 when compared to the corresponding edge counterparts.

Fig.S4 h *In vitro* cell viability assay of 101079 and 1051 edge or core GBM spheres treated with DMSO or AR42 at different concentrations.

Reviewer #3 (Remarks to the Author):

The addition of the data stemming from the addition of supplementary Fig. 5a in response to Major point 2 by Reviewer #2 improves the manuscript. However, there are several more points which should be addressed:

4) The new data in supplementary data 1f should be quantitated and stastically evaluated. I suspect that with the sample size (n=61 with 38 positive for both oligo and CD109 expression) divided into four groups (PN, CL, UK, and MES), will not under-powered and thus the data presented will not be significant. However, this needs to be stated so that these data can be put into appropriate context for the reader.

Reply: We apologies for the lack of explanation for this figure. The main purpose of that experiment was to demonstrate that Olig2⁺ and CD109⁺ cells can be found in all GBM subtypes and that edge-core signature can be observed in most, if not all, of GBM tumors. We added this information into the result section. In addition, we provided statistical analysis of these data. It demonstrates that MES tumors have a significantly lower expression of Olig2 and significantly higher expression of CD109 which is consistent with our previous report (*Cell Rep.* 2019, 26: 1893-1905).

Fig. S1 g Quantification of IHC staining intensity for Olig2 and CD109 of GBM samples obtained from 61 patients and related to different subtypes (proneural, classical, unknown and mesenchymal).

5) The term “HDAC1 signaling” is a bit of a misnomer as it is mainly a transducer through interactions with multiple other proteins and lysine acetylation of many non-histone proteins to impact a myriad of pathways. As most HDAC inhibitors affect transcription of approximately one third of the genome (with gene expression levels increased and decreased at almost equal frequencies), it is not unexpected that 1) CD109 expression is increased downstream of HDAC1 nor 2) CD109 promoter occupancy by HDAC1 and C/EBP β is low. However, the fact that it is not as novel as the authors’ posit, does not detract for their findings.

Reply: We apologize for the incorrect explanation. We added the following information into the discussion section: “Nonetheless, unanswered questions remain. It was previously shown that HDAC inhibition affects the expression of the substantial number of genes in the human genome and according to our data the level of the co-occupancy of HDAC1 and C/EBP β on the CD109 promoter detected by the ChIP experiment was rather low. Therefore, it is possible that HDAC1 may not be the main regulator of CD109 in core GBM cells. Rather, it may mediate the effect of the more specific regulator of CD109 expression that still has to be determined.”

6) The authors’ presentation of other HDAC inhibitors in clinical trials is a little misleading as well. There are a total of 10 HDACs in classes 1 and 2 and a single HDAC in class IV. Thus, it is difficult to claim AR-42 is significantly more specific than other HDAC inhibitors listed – particularly with the lack of established IC50s for AR-42 (lines 512-514).

Reply: We corrected this information in the discussion section: “There are several HDACs inhibitors in clinical trials that targets multiple different HDACs, like vorinostat, trichostatin A or panobinostat, targeting class I, II and IV HDACs. In our study, we used AR42 (class I and class II HDACs inhibitor) to decrease activity of HDAC1. According to our data, both AR42 and shRNA specifically targeting HDAC1 significantly decrease GBM growth both *in vitro* and *in vivo*. Thus, development and investigation of inhibitors more specific to target HDAC1 may potentially contribute to the future clinical treatment.”

Reviewer #4 (Remarks to the Author):

The overall concept promoted by the study, now better defined in the “graphical abstract” (Fig. 8F), is the existence of a distinct edge-localized GBM cell population which acts as recurrence-initiating cells (RICs) based on a set of properties of which some are cell-autonomous whereas others rely on paracrine signaling from post-surgery-remaining GBM core cells.

Also in the revised improved version the most interesting and translationally relevant findings are those that rely on the regionally defined GBM sub-populations. The summary below, and remaining major issues, are therefore mostly focusing on the experiments using those cell models and on the tissue analyses.

The edge RICs display certain gene-expression and GSEA profiles (Fig. 1 A, B, C, E) consistent with earlier published studies (Fig. 1 D). Furthermore sphere cultures of edge and core cells display, after brain injection, some properties consistent with their original counterparts including gene expression and pathway activation (Fig. 2 B, F,G). Edge cells show larger invasive capacity *in vivo* and *in vitro* (Fig. 2A, C). Furthermore, core cells are claimed to enhance some edge cell phenotypes, including invasion *in vivo* and *in vitro* (Fig. 2D,E), cell and tumor growth, IR

resistance and CD44 expression (Fig. 4). The paracrine capacity of edge cells are suggested to depend on HDAC1 based on shRNA and inhibitor experiments in studies that uses different endpoints including in vitro and in vivo cell growth (Fig. 6A-D) and molecular properties such as overall gene expression, c-myc expression, G2/M checkpoint (Fig. 6 E-G). Finally, a set of studies are shown to make the claim that C/EBPbeta-dependent production of CD109 by core cells are the HDAC1-dependent molecular mediators of the edge cell-derived paracrine capacities (Figs. 7 G, I, J; 8 C).

Major points

7) Fig 2A, B, C, F and G suggest stable cell autonomous differences between core and edge cells in analyses with endpoints including in vivo and in vitro invasion (2A, C) and c-myc, K-RAS, cell cycle status (2B, G). In contrast, experiments with the same models in Figs. 2D/E, 4, 6, 7 and 8 emphasize the strong dependence of edge cell phenotypes (cell growth, IR resistance and CD44 expression) on core cell-derived paracrine signaling. Notably, the cell autonomous c-myc and G2/M checkpoint phenotype of edge cells (Fig 2B, G) are in Fig. 6 E, F shown to be controlled by HDAC1-dependent paracrine signaling (Fig. 6E, F). This inconsistency should be resolved.

Reply: We thank reviewer for this important comment. To address this issue we added the following information into the discussion section: “Our data indicated that there is persistence of the spatial and phenotypic properties of GBM cells derived from tumor edge and core in alignment with that of the originating tumor’s regional identity. These findings raised the possibility that cell-intrinsic factors wrest control of spatial identity at an early time point of tumor development, leading to the appearance of stable cell autonomous differences between core- and edge-located GBM cells. Therefore, once GBM cell had acquired edge or core identity, it can be maintained even in the absence of tumor microenvironmental factors. However, in agreement with our previous observation (*Cancer Cell*. 2018, 34(1): 119-135), this cell autonomous phenotype can be affected by various paracrine signalings from another population of GBM cells. Altogether, our findings revealed an intricate interconnection between cell-intrinsic and microenvironmental factors that cooperatively determine the GBM spatial identity. We can propose that upon tumor growth, various factors such as low pH and hypoxia can trigger the acquisition of the core phenotype, which is then maintained by cell-intrinsic mechanisms. Importantly, these core cells can also disseminate some of their malignant properties to less aggressive GBM cells by producing a number of different extracellular signals.”

8) Authors are encouraged to more clearly highlight the fact that some edge cell phenotypes appear to be cell autonomous whereas other rely on edge cell-derived paracrine signaling. This should be clear in abstract and possibly also reflected in the title of the study.

Reply: We added the following sentences into the abstract: “To validate these data, we established regionally-derived models of GBM edge and core that retain the difference in their spatial identity in a cell autonomous manner and recapitulate paracrine signaling pathways, which may alter edge/core properties of the recipient cells.”

9) In this perspective it is notable that the infiltrative growth capacity (Fig. 2A, C) appear to be a cell autonomous property of edge cells. This raises some concerns regarding the therapeutic option, implied by the study, to block recurrences by interfering with paracrine edge/core cell signaling. This concern should be highlighted in the Discussion.

Reply: To address this issue we added the following information into the discussion section: “Thus, development and investigation of inhibitors more specific to HDAC1 may potentially contribute to the future clinical treatment. These new drugs may prevent acquisition of the aggressive and highly resistant core phenotype and therefore improve the efficiency of conventional chemo- and radiotherapy.”

10) Regarding the HDAC-C/EBPbeta-CD109 pathway, analyses still are missing for the regionally derived models regarding critical endpoints such as cell growth and IR-resistance.

Reply: To address this issue, we performed two sets of experiments using regionally-specified edge and core GBM cells from two different patients. First, we used ELISA to show that core, but not edge GBM cells secrete soluble CD109. Next, we added recombinant CD109 protein to the cultural medium of edge GBM cells and using FACS showed that purified CD109 can protect edge cells from radiation induced apoptosis. We added this information into the result section.

Fig.7 c Enzyme-linked immunosorbent assay (ELISA) for soluble CD109 in CM from 1051 and 101027 edge or core patient derived GBM spheres. **i** Flow cytometry analysis of caspase 3/7 activity and SYTOX staining in 1051 and 101027 edge or core spheres that were cultivated in a presence or absence of recombinant sCD109 for 3 days and subsequently irradiated with 8 Gy.

11) The study is still largely lacking correlative data from human tissue analyses to validate the HDAC-C/EBPbeta-CD109 pathway. Quality of study would be significantly improved if this pathway could be mapped in tissue sections using multiplex-antibody-profiling including pathway components and edge- and core-cell markers. Since authors imply the existence of edge cells in surgically removed tissue (Fig. 3) these studies could be possible to do in regular GBM specimens.

Reply: To address this issue we first used immunofluorescent staining and subsequent confocal microscopy to study distribution of Olig2⁺ cells and CD109⁺ cells within the tumor. As expected, we demonstrated that the tumor core is enriched for CD109⁺ cells, while tumor edge from the same patient contains more Olig2⁺ cells. Importantly, only a few cells showed simultaneous staining for both markers and it is possible that these cells appear due to the unspecific staining with both secondary antibodies rather than due to the simultaneous expression of both markers.

Fig.1 f Representative immunofluorescent staining of edge and core human GBM tissues for Olig2 (green), CD109 (red) and DNA (blue). Scale bar 50 μ m.

Next, we performed immunofluorescent staining with antibodies against HDAC1 and against Olig2 or CD109. According to our data there were no colocalization between HDAC1 and Olig2, while staining for HDAC1 and CD109 showed similar pattern (these two antibodies stained same cells, but antiHDAC1 stained nucleus while antiCD109 stained membranes). This result confirms our hypothesis that CD109⁺ core cells express high levels of HDAC1.

Fig.S5 h Representative immunofluorescent staining of human GBM tissues for HDAC1 (green), Olig2 (red) and DNA (blue) (upper) or for HDAC1 (green), CD109 (red) and DNA (blue) (lower). Scale bar 50 μ m.

Minor points

1. Findings of Fig. 3A are interesting and provides an important rational for the study. Possibly it can therefore be introduced as Fig. 1 A.

Reply: The first version of the manuscript had figure 3A as figure 1A, however, we moved it due to the request of other reviewers to make the story flow simpler and easier to follow.

2. Yellow star is missing in right panel of Fig. 2A.

3. The text below graphs of Fig. 2 G are not clear. They do not appear to indicate the groups that are compared in the GSEA analyses. If not; what do they indicate?

4. In Fig. 3D mouse pictures and graphs do not match; middle picture refers to top graph and top picture refers to middle graph

5. Text should be checked; e.g. “respectable tumor cell” (line 522) should probably be “resectable tumor cells”.

Reply: We apologize for these mistakes. We corrected it accordingly.

REVIEWER COMMENTS

Reviewer #2 (Remarks to the Author):

The authors have addressed my comments.

Reviewer #3 (Remarks to the Author):

General comments:

The manuscript is much improved and better substantiates 1) the communication between core and edge cells and 2) the link between these subtypes and tumor formation in vivo. In addition, the data explaining the HDAC1-C/EBP β -CD109 axis is put into better context. The involvement of CD109 is not surprising as the role of this molecule in glioblastoma has been documented in the literature as a biomarker (by some of these these authors and others e.g. PubMed 25724945, 30759398, 24069296). The upstream players in this pathway are largely part of more general transcription machinery and their association with the transcription of any gene is not surprising.

Specific comments:

1) The gene signatures in Figure 1 & Figure 1 supplemental are not carried forward carefully for data presented in subsequent figures. For instance, 1051, 1079, 1027, 1020, and 267 are included in supplemental figure 3a so that the reader can determine relative expression of CD44, Oligo2, CD133, and CD109. However, this is not the case for 1005, 20, 711, and 157 and only partially true for 101027 (it is in supplemental 3d but not 3a). This leads to the following questions moving through the figures:

a) Figure 2: How did the data from 101027 compare with that presented from 1051? Data from only one of two lines is presented. Why? From supplemental figure 3d, you might expect similar behavior between the two lines.

b) Figure 3: Work with 1501 continues in this figure and the defined 267 is added. But now work with 1005, 20, 711, and 157 is presented and there is not clear why some panels include some of these lines while others do not. There is no mention of 101027 and why it wasn't addressed.

c) Figure 4: The 101027 line is included with 1051 but not the others in figure 3. Why is 101027 included but not presented?

This is important as there is the expected heterogeneity in the gene signatures and without this information, the reader is at a loss to relate differences in gene signatures with characterized properties.

Given the number of authors and institutions, not all experiments may have been done on all patient derived lines. However, clear presentation of how and why only certain lines were used and their gene signatures should be presented. A supplementary table would suffice.

2) Is the raw RNA-seq data mentioned in figure 1b&c made available?

3) In figure 5a, FR901228 and FK228 are considered synonyms (see <https://pubchem.ncbi.nlm.nih.gov/compound/Romidepsin#section=Wikipedia>)

4) At points the authors clearly distinguish between cells and spheres with reference to passage number. A careful review of the language to make it consistent between figures is needed.

Reviewer #4 (Remarks to the Author):

Some improvements are recognized. There are still major issues that reduce the significance, stringency and relevance of the study.

1. The new Fig. 1 F is recognized as an addition. However, as reported now with only one case it

fails to demonstrate true regional differences in the distribution of the Olig2- and CD109-positive populations.

2. The revision of abstract is appreciated. However, it still fails to report that the potentially clinically relevant infiltrative growth capacity is a cell autonomous phenotype. In general abstract should be more specific in the description of which molecular and cellular phenotypes are shown to be cell autonomous and which are shown to be dependent on paracrine signaling.

3. The implications of therapeutic relevance of the proposed paracrine pathways, argued to be targeted in combination with radio- or chemo-therapy, remain unclear to me. These modalities are used post-surgery. It is not clear how targeting core-cell-derived signals can be useful since these are supposedly not present in the post-surgery situation. If this argument is correct the implications of therapeutic relevance of the paracrine pathways should be omitted from abstract.

Point by point reply to the reviewers' comments

Reviewer #3 (Remarks to the Author):

General comments:

The manuscript is much improved and better substantiates 1) the communication between core and edge cells and 2) the link between these subtypes and tumor formation in vivo. In addition, the data explaining the HDAC1-C/EBP β -CD109 axis is put into better context. The involvement of CD109 is not surprising as the role of this molecule in glioblastoma has been documented in the literature as a biomarker (by some of these these authors and others e.g. PubMed 25724945, 30759398, 24069296). The upstream players in this pathway are largely part of more general transcription machinery and their association with the transcription of any gene is not surprising.

Specific comments:

1) The gene signatures in Figure 1 & Figure 1 supplemental are not carried forward carefully for data presented in subsequent figures. For instance, 1051, 1079, 1027, 1020, and 267 are included in supplemental figure 3a so that the reader can determine relative expression of CD44, Oligo2, CD133, and CD109. However, this is not the case for 1005, 20, 711, and 157 and only partially true for 101027 (it is in supplemental 3d but not 3a). This leads to the following questions moving through the figures:

a) Figure 2: How did the data from 101027 compare with that presented from 1051? Data from only one of two lines is presented. Why? From supplemental figure 3d, you might expect similar behavior between the two lines.

b) Figure 3: Work with 1501 continues in this figure and the defined 267 is added. But now work with 1005, 20, 711, and 157 is presented and there is not clear why some panels include some of these lines while others do not. There is no mention of 101027 and why it wasn't addressed.

c) Figure 4: The 101027 line is included with 1051 but not the others in figure 3. Why is 101027 included but not presented?

This is important as there is the expected heterogeneity in the gene signatures and without this information, the reader is at a loss to relate differences in gene signatures with characterized properties.

Given the number of authors and institutions, not all experiments may have been done on all patient derived lines. However, clear presentation of how and why only certain lines were used and their gene signatures should be presented. A supplementary table would suffice..

Reply: We thank reviewer for this comment. One of the aims of this study was to demonstrate that the edge / core signature is general feature of a vast majority, if not all, of GBM tumors. Therefore, we tried to use as many patient-derived sphere lines as possible to test our hypothesis. To summarize the features of these cells, we included Table that contains a set of information about all GBM sphere lines used in the study (n=16). In short, we performed RNAseq for the 12 sphere lines and indicated CD44, CD109, CD133 and Olig2 expression in that Table. For the remaining 4 lines, expression of these representative markers was verified using RT-qPCR. According to the editor's request, we also added the information about the experiments which were performed with each sphere line. As can be seen from **Supplementary Table 1**, all experiments were performed

with at least 3 different sphere lines including at least 2 regionally-specified spheres lines. We apologize that we were unable to perform all experiments on all sphere lines.

Type	Sphere line	Origin	Rel. expression				RNaseq	qPCR	WB	Invasion		Coculture		IR sensitivity		CM cell growth	CM IR sensitivity	AR42 sensitivity		shHDAC	CD109 ELISA	rCD109 IR sensitivity	ChIP	
			CD133	Olig2	CD109	CD44				In vivo	In vitro	In vivo	In vitro	In vivo	In vitro			In vivo	In vitro					
Core	1051 core	UAB	1,08	0,56	1,23	1,61	YES	YES	YES	YES	YES	YES	YES	YES	YES	YES	YES	YES	YES	YES	YES	YES	YES	
	101027 core	UAB	0,03	0,78	1,82	1,17	YES	YES	YES	YES	YES	YES	YES	YES	YES	YES	YES	YES	YES	YES	YES	YES	YES	
Edge	1051 edge	UAB	2,77	1,04	0,29	0,24	YES	YES	YES	YES	YES	YES	YES	YES	YES	YES	YES	YES	YES	YES	YES	YES	YES	
	101027 edge	UAB	0,12	1,62	0,67	0,97	YES	YES	YES	YES	YES	YES	YES	YES	YES	YES	YES	YES	YES	YES	YES	YES	YES	
Core-like	267	MDA	0,10	0,01	5,25	1,13	YES	YES			YES	YES	YES	YES	YES	YES	YES	YES	YES	YES	YES	YES	YES	
	1005	UAB	0,71	1,08	0,93	0,03	YES	YES			YES	YES	YES	YES	YES	YES	YES	YES	YES	YES	YES	YES	YES	
	1020	UAB	0,00	0,00	0,69	1,10	YES	YES																
	28	UAB	0,01	0,01	0,85	5,36	YES	YES				YES								YES				
	339	UCLA	Low		High		YES	YES																
20	MDA	Low		High		YES	YES								YES			YES		YES				
Edge-like	157	UCLA	2,34	2,24	0,00	0,12	YES	YES	YES		YES	YES	YES	YES	YES	YES	YES	YES	YES	YES	YES	YES	YES	YES
	1051	UAB	2,21	1,64	0,01	0,03	YES	YES			YES	YES	YES	YES	YES	YES	YES	YES	YES	YES	YES	YES	YES	YES
	1079	UAB	1,43	1,89	0,18	0,19	YES	YES					YES										YES	YES
	1027	UAB	1,21	1,13	0,08	0,04	YES	YES																
	408	UCLA	High		Low		YES	YES												YES				
711	MDA	High		Low		YES	YES	YES					YES			YES		YES						

Supplementary Table 1. List of all GBM sphere lines and the experiments in which the indicated spheres were used in the study. Expression of key edge/core markers is indicated (high resolution table is included in supplementary materials).

2) Is the raw RNA-seq data mentioned in figure 1b&c made available?

Reply: RNAseq data of 12 GBM sphere lines as well as all RNAseq data from Fig.1 (9 tissue samples) will be deposited at Gene Expression Omnibus database after acceptance of the manuscript.

3) In figure 5a, FR901228 and FK228 are considered synonyms (see <https://pubchem.ncbi.nlm.nih.gov/compound/Romidepsin#section=Wikipedia>).

Reply: We apologize for this mistake. We corrected the table accordingly.

4) At points the authors clearly distinguish between cells and spheres with reference to passage number. A careful review of the language to make it consistent between figures is needed.

Reply: We thank the reviewer to point us to this inconsistency. We have made necessary edits throughout the manuscript to make it more suitable for publication.

Reviewer #4 (Remarks to the Author):

Some improvements are recognized. There are still major issues that reduces the significance, stringency and relevance of the study.

5) The new Fig. 1 F is recognized as an addition. However, as reported now with only one case it fails to demonstrate true regional differences in the distribution of the Olig2- and CD109-positive populations.

Reply: We apologize for the lack of samples. We performed IHC with antibodies against Olig2 and CD109 using 3 more paired edge/core samples. Figure S1C shows that edge samples are enriched with Olig2+ cells, while core samples show higher expression of CD109. These observation is consistent with what we have shown in the previous version of our manuscript providing further confirmation.

Fig. S1 c IHC staining of paired edge and core tumor tissues for Olig2 (upper) and CD109 (lower). Scale bar 200 μ m.

6) The revision of abstract is appreciated. However, it still fails to report that the potentially clinically relevant infiltrative growth capacity is a cell autonomous phenotype. In general abstract should be more specific in the description of which molecular and cellular phenotypes are shown to be cell autonomous and which are shown to be dependent on paracrine signaling.

Reply: We corrected Abstract accordingly: “Intratumor spatial heterogeneity facilitates therapeutic resistance in glioblastoma (GBM). Nonetheless, understanding of spatial heterogeneity

in GBM is limited to the resectable tumor core lesion. In sharp contrast, the seeds for tumor recurrence, termed recurrence-initiating cells, reside in the surgically unresectable tumor edge. In this study, stratification of GBM to core and edge according to the regional characteristics demonstrated clinically relevant surgical sequelae. Edge cells showed a higher capacity for infiltrative growth, while core cells demonstrated greater therapy resistance. Investigation of intercellular signaling between these two cell populations uncovered the presence of paracrine crosstalk from tumor core that provokes malignancy and therapy resistance of edge cells. These phenotypic alterations were initiated by HDAC1 signaling in GBM core cells which subsequently affected edge cells by secreting the soluble form of CD109 protein. To validate these data, we established regionally derived models of GBM edge and core that retained the difference in their spatial identity in a cell autonomous manner and recapitulated paracrine signaling pathways. Collectively, this study reveals the role of intracellular communication between regionally different populations of GBM cells in tumor recurrence.”

7) The implications of therapeutic relevance of the proposed paracrine pathways, argued to be targeted in combination with radio- or chemo-therapy, remain unclear to me. These modalities are used post-surgery. It is not clear how targeting core-cell-derived signals can be useful since these are supposedly not present in the post-surgery situation. If this argument is correct the implications of therapeutic relevance of the paracrine pathways should be omitted from abstract.

Reply: We would respectfully point out that according to the data presented on Fig.3A, more than 2/3 of GBM patients undergo -incomplete surgical resection with residual core cells identifiable by MRI and these patients are characterized by much worse survival. Therefore, targeting core-cell-derived signals is highly likely to be beneficial for these patients. In addition, even in case of complete resection, remaining edge GBM cells eventually establish lethal tumor core as recurrence due to the newly established edge-core transition with unknown intercellular signaling. Having said that, upon the Editor’s and Reviewer’s requests, we have modified the Abstract.

REVIEWERS' COMMENTS:

Reviewer #3 (Remarks to the Author):

The authors have adequately addressed my concerns.

Reviewer #4 (Remarks to the Author):

The addition of novel data and modifications of abstract are appreciated and recognized.

Concerning last point about therapeutic relevance, I accept the argument from authors and withdraw the recommendation to take away from abstract the implications of therapeutic relevance of findings. Authors are welcome to keep that part phrased in a careful manner.

Finally, authors can consider in their Discussion potential relationships between the "edge cells" and the recently proposed "outer radial glia (oRG)" implied in invasive growth of GBM (Bhaduri et al., Cell Stem cell, 2020).